# LoRA-Mixer: Coordinate Modular LoRA Experts Through Serial Attention Routing

**Wenbing Li, Zikai Song, Hang Zhou, Yunyao Zhang, Junqing Yu, Wei Yang** *

Huazhong University of Science and Technology
`{d202581843}@hust.edu.cn`

## Abstract

Recent attempts to combine low-rank adaptation (LoRA) with mixture-of-experts (MoE) for multi-task adaptation of Large Language Models (LLMs) often replace whole attention/FFN layers with switch experts or append parallel expert branches, undermining parameter efficiency and limiting task specialization. We introduce **LoRA-Mixer**, a modular MoE framework that routes task-specific LoRA experts into the core projection matrices of the attention module (input/output linear layers), rather than primarily targeting FFN blocks. The design delivers fine-grained token-level specialization by fully exploiting the attention mechanism, while remaining drop-in compatible with Transformers and state-space models (SSMs) as the linear projection layers are ubiquitous. To train robust routers from limited data while promoting stable, selective decisions and high expert reuse, **LoRA-Mixer** employs an adaptive **Routing Specialization Loss (RSL)** that jointly enforces global load balance and input-aware specialization via an entropy-shaping objective. The framework supports two regimes: (i) joint optimization of adapters and router with a differentiable hard–soft top-$k$ routing scheme, and (ii) plug-and-play routing over frozen, pre-trained LoRA modules sourced from public repositories. Across 15 benchmarks—including MedQA, GSM8K, HumanEval, and GLUE—RSL-optimized LoRA-Mixer outperforms state-of-the-art routing and LoRA-MoE baselines while using 48% of their trainable parameters, with gains of +3.79%, +2.90%, and +3.95% on GSM8K, CoLA, and ARC-C, respectively. Cross-model transfer and adapter reuse experiments further demonstrate the approach's versatility and data efficiency. Our code can be obtained at https://github.com/hustcselwb/LoRA-Mixer.

## 1 Introduction.

Large Language Models (LLMs) have achieved unprecedented proficiency in general-purpose reasoning and generation, yet their adaptation to specialized downstream domains remains computationally prohibitive for full-scale fine-tuning Brown et al. (2020); Touvron et al. (2023). To mitigate these resource demands, parameter-efficient fine-tuning (PEFT) methods Li & Liang (2021); Zaken et al. (2021); Liu et al. (2022); Houlsby et al. (2019); Liu et al. (2021) have emerged as a scalable paradigm. Among them, Low-Rank Adaptation (LoRA) Hu et al. (2022); Tian et al. (2024) has demonstrated particular efficacy, operating through low-rank decomposition of updates to the pre-trained weights—enabling efficient tuning with minimal parameter overhead. Recent work has explored modularly composing independently trained LoRA modules as a promising strategy for multitask adaptation; however, naive composition can result in interference between task-specific subspaces, limiting their synergistic potential Huang et al. (2023); Wu et al. (2024). This limitation has motivated exploration of mixture-of-experts (MoE) architecture Shazeer et al. (2017); Fedus et al. (2022), which treat each task-specific LoRA as an expert and sparsely activate and fuse the experts. Recent studies demonstrate promising directions in hybrid LoRA-MoE frameworks Wu et al. (2024); Li et al. (2024a); Huang et al. (2023); Zhao et al. (2024); Liao et al. (2025); Gao et al. (2024); Chen et al. (2024); Dou et al. (2023), aiming to enhance model performance on complex tasks across multi-domain datasets while preserving the parameter efficiency of fine-tuning.

---

*denotes the corresponding author.

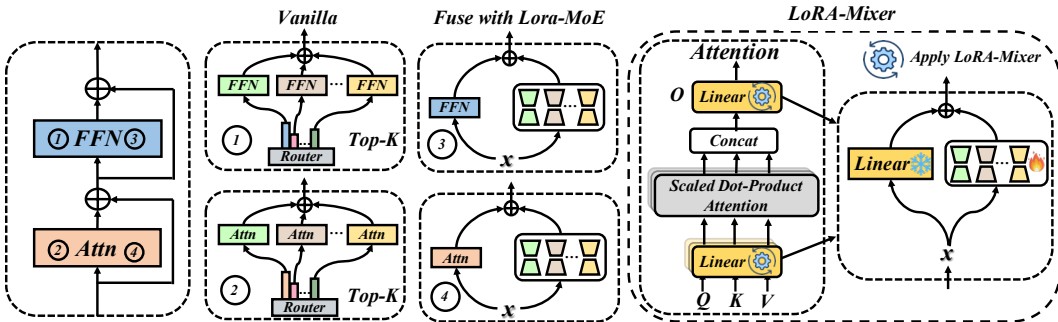

Figure 1: LoRA and MoE integration methods. (1) and (2): replace the attention or FNNs with switch experts; (3) and (4): introduce LoRA experts branches in parallel with the attention or feedforward layers, and fuse the output into the main branch. Our **LoRA-Mixer** (right) applies mixture of LoRA experts to the projection layers, which can effectively leverages the attention mechanism.

The central challenge in composing pre-trained LoRAs is to realize synergy—improving performance across constituent tasks—without inflating training cost or erasing task-specific inductive biases. Existing LoRA–MoE integrations largely follow two patterns: (i) replacing attention or FFN blocks with LoRA-based switch experts in a standard MoE setup Li et al. (2024a); Chen et al. (2024); Fedus et al. (2022); Shazeer et al. (2017); or (ii) attaching parallel LoRA branches whose outputs are fused back into the main path Wu et al. (2024); Huang et al. (2023) (Fig. 1). Despite encouraging results Wu et al. (2024); Liao et al. (2025); Krajewski et al. (2024); Zhang et al. (2019); Li et al. (2022); Gross et al. (2017), switch-style designs typically require joint training of all experts, increasing data demands and hindering modular reuse and transfer of off-the-shelf LoRAs. Parallel-branch schemes sidestep native attention or state-transition pathways, yielding shallow output fusion and weak integration. Moreover, common auxiliary routing losses emphasize uniform load balancing, suppressing input- and task-aware specialization Li et al. (2024a). These issues motivate a plug-and-play, architecture-agnostic alternative—compatible with both Transformers and state-space models (SSMs) Gu & Dao (2023)—that learns discriminative routing with minimal compute and data while maximizing reuse of independently trained LoRA modules.

In response, we introduce **LoRA-Mixer**, a novel framework designed to efficiently synergize multiple pre-trained LoRA modules by treating them as dynamic, pluggable memory cells. LoRA-Mixer equips the linear projection layers of the original model with mixed LoRA experts, enabling these experts to directly leverage the core attention or state-transition mechanisms. To further enhance efficiency and maintain routing effectiveness, **LoRA-Mixer** adopts a novel Router Specialization Balancing Loss (**RSL**) to align routing decisions with token-level expert usage, maintaining moderate entropy to encourage exploratory behavior. During inference, we employ sparse top-$K$ fusion, effectively balancing computational cost and scalability without compromising expert selectivity. LoRA-Mixer supports LoRA modules sourced from external repositories, independently trained, or jointly trained through hard routing strategies, allowing seamless plug-and-play usage across various tasks and domains. Importantly, our method significantly reduces the necessity for training data or extensive re-adaptation, requiring only minimal additional data to effectively train the routing mechanism. Consequently, LoRA-Mixer is particularly suitable for constructing large-scale, modular language models characterized by task-specific memory, computational efficiency, and strong transferability. Extensive evaluation on 15 benchmark datasets (MedQA, GLUE, GSM8K, ARC-E, ARC-C, HumanEval, PIQA, HellaSwag, and BoolQ) demonstrates that integrating LoRA-Mixer significantly improves model performance across all evaluated tasks. With only 48% of the parameters of existing methods, the RSL-optimized LoRA-Mixer outperforms the state-of-the-art routing and LoRA-MoE baseline models, achieving improvements of +3.79%, +2.90%, and +3.95% on GSM8K, CoLA, and ARC-C, respectively. Cross-model transfer and adaptive module reuse experiments further demonstrate the method's versatility and data efficiency.

## 2 RELATED WORK.

**PEFT For LLMs** Low-rank adaptation (LoRA) Hayou et al. (2024); Zhou et al. (2024); Hu et al. (2022); Xu et al. (2024); Sheng et al. (2023); Biderman et al. (2024); Li et al. (2023); Zeng et al. (2025);

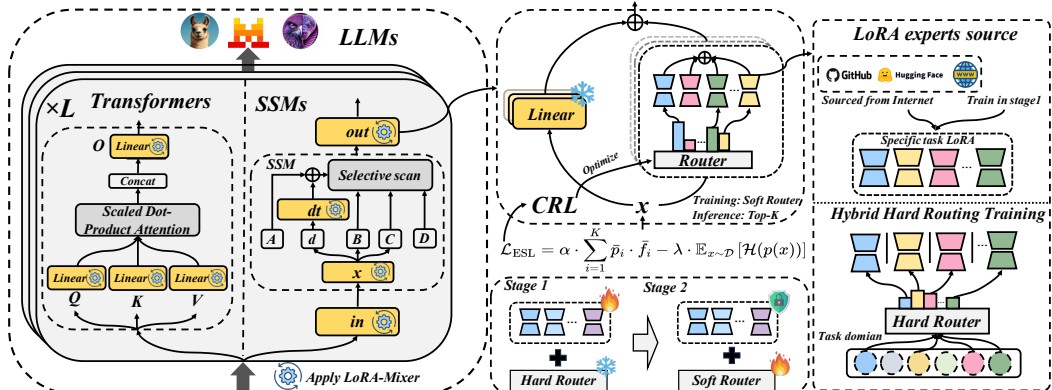

Figure 2: The overall architecture of LoRA-Mixer. LoRA-Mixer is applied to the linear projection layers in serial with the Attention and SSM modules and support all major LLM structures. LoRA-Mixer reueses the LoRA experts sourced from Internet, trained individually or jointly trained using hard routing. The routing training is guided by RSL loss for balancing experts loads and specificity.

Lin et al. (2025a;b) effectively fine-tunes large models by learning a low-rank matrix and freezing the original weights. While effective for a single task, its task-specific nature limits generalization. Recent work combines LoRA with mixture of experts (MoE) Li et al. (2024a); Feng et al. (2025a); Wu et al. (2024); Ouyang et al. (2025); Feng et al. (2025b); Huang et al. (2023); Zhao et al. (2024); Song et al. (2022; 2023); Hu et al. (2025) to achieve dynamic adaptation. For example, MixLoRA Li et al. (2024a) uses LoRA experts for top-k routing in FFN, improving multi-task performance but suffering from gradient entanglement issues. MoLE Wu et al. (2024) combines LoRA layers via gating but lacks sparse routing. LoraHub Huang et al. (2023) performs gradient-free few-shot combination of LoRA modules for unseen tasks but struggles with complex semantics due to lack of gradient optimization and dynamic routing. Other methods Li et al. (2025); Huang et al. (2025); Yang et al. (2024); Chen et al. (2023); Luo et al. (2024); Li et al. (2024b); Feng et al. (2025c) explore flexible routing mechanisms to improve the model's adaptability. However, these methods usually introduce additional routing networks or optimization targets, resulting in instability during training, limiting their application in actual multi-task or low-resource scenarios.

**Mixture of Experts** In recent years, the mixture of experts (MoE) architecture has attracted much attention as a promising LLM expansion paradigm. By selectively activating a subset of expert modules for each input, MoE allows the model to scale capacity without linearly increasing the amount of computation. As a result, more and more large models have adopted MoE, including GLaMDu et al. (2022), Switch TransformersFedus et al. (2022); Song et al. (2025); Zhang et al. (2025a;b); Ye et al. (2025), and the recent DeepSeek seriesLiu et al. (2024); Guo et al. (2025a). These advances indicate that MoE is becoming a mainstream architectural trend in the development of next-generation base models. Among them, GLaMDu et al. (2022) and Switch TransformerFedus et al. (2022) build a mixture of experts (MoE) model in the FFN module, and use a sparse activation mixed expert architecture to expand the model capacity and achieve better performance. LLaVA-MoLEChen et al. (2024) uses a top-1 strategy to route tokens to domain-specific expert models, thereby alleviating data conflicts and achieving continuous performance improvements over the ordinary LoRA baseline. LoRAMoEDou et al. (2023) uses routers to integrate LoRA experts while retaining general knowledge. HMoRALiao et al. (2025) combines the layered fine-tuning methods of MoE and LoRA, and gradually switches the routing strategy as the number of layers increases. MoLAGao et al. (2024) assigns different numbers of experts at different levels, proving that deeper layers often require more experts. Notably, we only utilize hybrid LoRA experts in the core projection layer, resulting in extremely high parameter efficiency. By applying RSL to LoRA-Mixer, we can efficiently reuse LoRA modules and require minimal training data, saving computational resources while expanding model capacity and improving generalization.

## 3  METHOD.

In this section, we introduce **LoRA-Mixer**, a flexible and pluggable Mixture of Experts (MoE) framework for combining LoRA experts from multiple LLMs.

## 3.1 PRELIMINARIES

**Mixture-of-Experts** is a sparse neural architecture where each input token is processed by a small subset of expert networks. Given $K$ experts and a router that produces a score vector $G(\mathbf{x}) \in \mathbb{R}^K$, a softmax is applied to obtain the routing distribution:

$$p_i(\mathbf{x}) = \frac{\exp(G_i(\mathbf{x}))}{\sum_{j=1}^{K} \exp(G_j(\mathbf{x}))}, \quad i = 1, \ldots, K. \tag{1}$$

The top-$k$ experts are selected based on $p_i(x)$, and the final output is computed as a weighted sum over their outputs:

$$\text{MoE}(\mathbf{x}) = \sum_{i=1}^{K} \mathbb{I}[i \in \text{TopK}(p(\mathbf{x}))] \cdot p_i(\mathbf{x}) \cdot \text{Expert}_i(\mathbf{x}) \tag{2}$$

This design allows MoE to reduce computational cost while enabling experts to specialize on different input patterns.

**Auxiliary loss.** To encourage balanced expert utilization, MoE training introduces an auxiliary loss. We define the expected routing probability and top-1 usage as

$$\bar{p}_i = \mathbb{E}_{x \sim \mathcal{D}}[p_i(x)], \qquad \bar{f}_i = \mathbb{E}_{x \sim \mathcal{D}}\big[\mathbb{I}(i = \arg\max_j p_j(x))\big], \qquad L_{\text{aux}} = \alpha \sum_{i=1}^{K} \bar{p}_i\, \bar{f}_i. \tag{3}$$

where $\bar{p}_i$ denotes the average routing intention and $\bar{f}_i$ the empirical usage. We demonstrate in Appendix A.17 why the auxiliary loss leads to over-averaging.

## 3.2 LoRA-MIXER FOR COMPOSITING LoRAs

Combining independently trained LoRA modules for multi-task adaptation provides a promising approach to provide LLMs with cross-domain composition capabilities. For example, we can fuse mathematics- and medicine-specific LoRA to enable LLMs to have both stronger mathematical reasoning capabilities and medical-specific knowledge to solve complex cross-domain queries.

Our proposed **LoRA-Mixer** implements this mechanism by treating each pre-trained LoRA module as an expert and learning a routing function $\mathcal{F}_{\text{route}}$ that dynamically fuses these experts based on the input semantics. The routing mechanism is lightweight and data-efficient, and can achieve task awareness with only a small amount of additional training. LoRA-Mixer uses a set of $E$ low-rank experts and a router $\alpha(x) \in \mathbb{R}^E$ to enhance the pre-trained projection matrix $W \in \mathbb{R}^{d_{\text{out}} \times d_{\text{in}}}$. Each expert is parameterized as $\Delta W^{(e)} = A^{(e)} B^{(e)}$, where $A^{(e)} \in \mathbb{R}^{d_{\text{out}} \times r}$ and $B^{(e)} \in \mathbb{R}^{r \times d_{\text{in}}}$. The output of LoRA-Mixer is:

$$\mathbf{y} = W\mathbf{x} + \mathcal{F}_{\text{route}}\left(\left\{\alpha_e(\mathbf{x}) \cdot \Delta W^{(e)}\mathbf{x}\right\}_{e=1}^{E}\right) \tag{4}$$

where, $\mathcal{F}_{\text{route}}(\cdot)$ represents the routing function output by the fusion expert. The output will be passed to the subsequent attention module or state-space module, enabling it to directly influence the core representation learning path. This strategy ensures that LoRA-Mixer acts at the most expressive point of the model - the projection layer - without disrupting the underlying architecture.

**LoRA Experts Acquirement.** Our proposed LoRA-Mixer framework is highly flexible and supports the integration of LoRA modules from diverse sources. In one common scenario, users may download pre-trained LoRA adapters from public repositories such as LoRAHub Huang et al. (2023), which currently hosts 196 high-quality LoRA modules across a wide range of domains. These can be directly composed using LoRA-Mixer with minimal additional data. Alternatively, users may independently train domain-specific LoRA modules tailored to their own datasets. For scenarios requiring joint training of multiple LoRA modules on a heterogeneous, labeled dataset, LoRA-Mixer further supports a hard-routing strategy. Specifically, we fix the routing module and apply a deterministic routing scheme based on known domain labels. Given a domain ID $d \in \{1, \ldots, K\}$ associated with each training instance, all tokens within the sample are routed exclusively to expert $d$. This design enables efficient joint optimization while maintaining expert modularity. Collectively, these capabilities make LoRA-Mixer a versatile and scalable framework for composing heterogeneous LoRA modules. The overall architecture is illustrated in Figure 2.

## 3.3 Specialization Balance Loss for Routing Optimization

The next step is to optimize the expert routers. While previous research has introduced auxiliary loss functions to align the average gating score with expert utilization, thereby promoting load balancing, we observed that this approach overemphasizes consistency, resulting in an overly balanced expert distribution. In this case, all experts are forced to be used equally, regardless of the input semantics. This hinders effective routing and generally requires more training data.

We propose a new perspective: viewing the router as an information bottleneck that determines the degree to which semantic differences between tokens are preserved or compressed. From this perspective, the entropy of the routing distribution quantifies the uncertainty of the router. Therefore, load balancing and specialization selection appear to be in conflict. To achieve a balance between the two, ensuring balanced expert load and input-aware routing, we propose an improved optimization objective, called Routing Specialization Balance Loss (RSL). RSL addresses this issue by unifying the global budget and local selectivity into a single objective. Specifically, RSL enhances the auxiliary loss with an entropy regularization term, improving the accuracy of expert selection by suppressing overly flat distributions. Formally, let $\bar{p}_i$ denote the average soft routing score (across tokens) and $\bar{f}_i$ denote the normalized score assigned to the token of expert $i$ in the first $k$ routes. The RSL loss function is defined as:

$$\mathcal{L}_{\text{RSL}} = \alpha \cdot \sum_{i=1}^{K} \bar{p}_i \cdot \bar{f}_i - \lambda \cdot \mathbb{E}_{x \sim \mathcal{D}} \left[ \mathcal{H}(p(\mathbf{x})) \right], \tag{5}$$

Where $\alpha$ controls the strength of the equilibrium consistency term, $\lambda$ is a small positive coefficient of the entropy regularization term:

$$\mathcal{H}(p(\mathbf{x})) = -\sum_i p_i(\mathbf{x}) \log p_i(\mathbf{x}) \tag{6}$$

The design principles behind RSL are: 1) From an information bottleneck perspective, minimizing $\mathcal{H}(p(x))$ reduces token-conditional uncertainty under a fixed global load, directly promoting specialization without disrupting the balance. 2) The entropy term acts as a curvature provider in the routing simplex, resulting in strong convexity and well-conditioned optimization (see A.1); this is precisely where traditional auxiliary loss functions are weak. 3) The coefficient $\lambda$ becomes an interpretable knob that trades off global fairness and local specialization, rather than simply a regularization weight.

Specifically, the entropy term introduces token-specific entropy. We derive the entropy gradient with respect to the routing score $p_i(x)$:

$$\frac{\partial \mathcal{H}(p(x))}{\partial p_i(x)} = -\log p_i(x) - 1, \quad \text{subject to} \quad \sum_{i=1}^{K} p_i(x) = 1, \tag{7}$$

$$= -\log p_i(x) - 1 + \mu, \tag{8}$$

where $\mu$ is the Lagrange multiplier due to the simplex constraint. Therefore, the total gradient of the RSL loss becomes:

$$\nabla_{p_i(x)} \mathcal{L}_{\text{RSL}} = \alpha \cdot \frac{\partial \bar{p}_i}{\partial p_i(x)} \cdot \bar{f}_i + \lambda(\log p_i(x) + 1 - \mu). \tag{9}$$

This demonstrates that RSL introduces a token-level gradient signal via $\log p_i(x)$, unlike the auxiliary loss, which only propagates global gradients. Specifically, the $\log p_i(x)$ term provides a token-level signal that amplifies the input-aware variance in the expert's choices, thereby counteracting the bias towards consistency in global balance. To quantify input-aware routing, we define:

$$\text{Var}_{x \sim \mathcal{D}}(p(x)) := \mathbb{E}_x \left[ \|p(x) - \bar{p}\|^2 \right]. \tag{10}$$

We call routing token-aware if $\text{Var}(p(x)) > \epsilon$ and $\epsilon > 0$. The auxiliary loss function tends to reduce this variance (driving uniform routing), while RSL encourages high variance and peaked distributions that are consistent with the input semantics. In addition to the above proofs, we provide **convergence analysis** and **generalization bound** proofs in the Appendix A.1 A.2. The former introduces strong convexity on the product simplex and proves that the added entropy term can maintain stable optimization dynamics; the latter shows that entropy regularization reduces the assumed complexity of the router and improves robustness in low-data environments.

**Routing Optimization.** After LoRA is ready, we soft-train the router. To prevent expert knowledge from being contaminated, we introduce a regularization term to penalize deviations from the previously learned expert parameters. Let the first-stage parameters of expert $i$ be $\theta_i^{(0)}$ and the current parameters be $\theta_i$. We define the regularization term as:

$$\mathcal{L}_{\text{preserve}} = \beta \cdot \sum_{i \in \mathcal{C}} \left\| \theta_i - \theta_i^{(0)} \right\|^2 = \beta \cdot \sum_{i \in \mathcal{C}} \left\| \Delta \theta_i \right\|^2, \tag{11}$$

where $\mathcal{C}$ represents the set of constrained experts and $\beta$ controls the regularization strength. This regularization term constrains sensitive experts to maintain their original knowledge while allowing other experts to flexibly adjust, supporting multi-expert learning for complex tasks.

To ensure effective gradients and stable optimization during joint training, we employ soft expert fusion: the routers output a softmax score $\mathbf{p}_{b,t} \in \mathbb{R}^K$, achieving differentiable LoRA hybridization. While this approach provides stable gradients, combining it with an auxiliary loss results in equal activations among experts, inhibiting specialization. To address this, we introduce RSL, which promotes expert differentiation while maintaining load balancing and enables efficient reuse of LoRA with minimal data. The total loss during joint training is:

$$\mathcal{L}_{\text{total}} = \mathcal{L}_{\text{task}} + \alpha \cdot \mathcal{L}_{\text{RSL}} + \beta \cdot \sum_{i \in \mathcal{C}} \left\| \theta_i - \theta_i^{(0)} \right\|^2, \tag{12}$$

where $\mathcal{L}_{\text{task}}$ denotes the standard task loss (e.g., cross-entropy loss). $\mathcal{L}_{\text{RSL}}$ is the **routing specialization balance loss**, which is used to promote consistency between routing scores and actual token assignments. $\alpha$ is a weighting factor for RSL. $\mathcal{L}_{\text{preserve}}$ is the expert regularization loss described above, scaled by $\beta$. We provide an exploration of how to choose hyperparameters in Appendix A.8.

## 4 EXPERIMENT.

In this section, we conduct extensive experiments on 15 datasets across five domains (MedQA, commonsense reasoning, NLP, mathematics, and coding).

### 4.1 EXPERIMENTAL SETUP

**Datasets** We tested RSL and LoRA-Mixer on 15 public benchmarks: MedicalQA, ARC-E, ARC-C, GSM8K, GLUE, PIQA, BoolQ, HumanEval, and HellaSwag. These datasets cover five different domains. For detailed dataset information, please refer Appendix 11.

**Baseline** We selected LLaMA3-8B, Mistral-7B, and Falcon-Mamba-7B as the basemodel. LLaMA3-8B and Mistral-7B are Transformer architectures, while Falcon-Mamba is a pure SSM architecture. We compare LoRA-Mixer and RSL with other state-of-the-art methods, including MoLE Wu et al. (2024), MixLoRA Li et al. (2024a), LoraHub Huang et al. (2023), LoRA-LEGO Zhao et al. (2024) and PHATGOOSE Muqeeth et al. (2024). We also select strong baselines that specifically optimize auxiliary losses to demonstrate the effectiveness of RSL, including GMoE Bai et al. (2024), AESL Guo et al. (2025b), and DsMoE Pan et al. (2024). It is worth noting that MoLE and MixLoRA also optimize auxiliary losses. For parameter, training and inference analysis, please refer to A.4 A.7.

**Metrics** For HumanEval, we use the $Pass@1$ metric. Considering the domain-specific freedom and rigor required by the Medical-QA dataset, we use DeepSeek-R1 for evaluation. For the remaining tasks, we use ACC as the metric. To reduce randomness, all experiments are run three times and the average reported.

### 4.2 COMPARISONS

Table 1: Performance of three base models, Falcon-Mamba-7B, Mistral-7B, and LLaMA3-8B, on seven benchmarks.

| Base Model | Medical | CoLA | SST2 | GSM8K | ARC-E | ARC-C | HumanEval |
|---|---|---|---|---|---|---|---|
| Falcon-Mamba-7B | 73.67 | 82.42 | 92.81 | 52.54 | 77.61 | 68.78 | 29.29 |
| Mistral-7B | 66.32 | 71.21 | 85.24 | 40.83 | 80.00 | 61.50 | 27.95 |
| LLaMA3-8B | 78.47 | 79.14 | 93.12 | 57.92 | 88.45 | 78.65 | 52.44 |

Table1 shows the performance indicators of the three basemodels on various tasks. We compare our approach with the state-of-the-art methods on seven datasets. The experimental results are shown in

Table 2. Our approach outperforms the baselines on most tasks. For the Falcon-Mamba dataset, our approach significantly outperforms the baselines on all tasks. For model details, see Appendix A.11.

We use the LLaMA2-7B from the LoRA-LEGO paper as the basemodel and conduct experiments on the four tasks: CoLA, SST2, MRPC, and RTE. The LoRA configuration uses $r = 6$ and $\alpha = 12$. The experimental results are shown in Table 4. As can be seen, our method outperforms LoRA-LEGO on three of the four tasks.

Table 2: Comparison of our LoRA-Mixer with LoRAHub, MoLE, and Mix-LoRA across seven tasks (best scores in bold). Note that MixLoRA is excluded from Falcon-Mamba due to its Transformer-specific design.

| Metho | Medical | CoLA | SST2 | GSM8K | ARC-E | ARC-C | HumanEval |
|---|---|---|---|---|---|---|---|
| *Falcon-Mamba (7B)* | | | | | | | |
| LoRAHub | 70.14 | 81.11 | 93.35 | 51.64 | 81.16 | 72.37 | 30.68 |
| MOLE | 74.51 | 84.77 | 94.22 | 54.28 | 83.46 | 76.61 | 33.57 |
| LoRA | 77.26 | 85.62 | 95.07 | 56.27 | 85.68 | 76.51 | 33.54 |
| LoRA-Mixer (ours) | **78.01** | **85.91** | **95.76** | **57.87** | **86.87** | **77.19** | **35.37** |
| *Mistral (7B)* | | | | | | | |
| LoRAHub | 69.17 | 75.73 | 90.21 | 44.94 | 81.14 | 69.21 | 32.60 |
| MOLE | 71.07 | 78.51 | 94.17 | 45.31 | 85.68 | 68.77 | 35.37 |
| MixLoRA | 69.74 | 78.61 | 93.44 | 45.50 | 85.42 | 69.15 | 33.80 |
| LoRA | 70.33 | 79.19 | 93.58 | **46.67** | 86.66 | 70.53 | 35.31 |
| LoRA-Mixer (ours) | **71.25** | **82.17** | **95.16** | 46.48 | **87.87** | **71.22** | **36.76** |
| *LLaMA-3 (8B)* | | | | | | | |
| LoRAHub | 78.11 | 79.84 | 92.77 | 59.10 | 87.13 | 80.14 | 52.83 |
| MOLE | 78.43 | 81.37 | 94.18 | 63.81 | 88.15 | 81.77 | 55.87 |
| MixLoRA | 79.87 | 80.67 | 94.22 | 64.44 | 88.70 | 82.90 | 55.49 |
| LoRA | 81.09 | 81.50 | 95.30 | 65.14 | 89.59 | 82.15 | 55.61 |
| LoRA-Mixer(ours) | **81.55** | **82.22** | **95.41** | **65.53** | **89.88** | **83.24** | **57.32** |

Table 3: Evaluation of LoRA-Mixer on LoRAs sourced from Internet on five GLUE tasks, the base model is Flan-T5Chung et al. (2024).

| Method | SST-2 | CoLA | MRPC | RTE | QQP |
|---|---|---|---|---|---|
| Flan-T5 | 94.01 | 74.21 | 79.90 | 80.08 | 82.32 |
| LoRA | 94.50 | 80.54 | 83.76 | 83.47 | **85.55** |
| LoRA-Mixer | **95.07** | **82.14** | **85.15** | **85.31** | 84.75 |

Table 4: Comparison of LoRA-Mixer and LoRA-LEGO. Results for LoRA-LEGO are from its paper.

| Method | CoLA | SST-2 | MRPC | RTE |
|---|---|---|---|---|
| LoRA | 61.63 | 75.74 | 68.00 | 52.22 |
| LEGO | 55.48 | 73.22 | 66.00 | **71.85** |
| LoRA-Mixer | **64.60** | **80.31** | **72.24** | 61.47 |

Considering that Mistral-7B and LLaMA3-8B have the same architecture, we directly migrate the parameters trained on Mistral-7B to LLaMA3-8B without any fine-tuning and adaptation, and conduct experiments on three datasets: ARC-E, ARC-C, and GSM8K. The results are shown in Table 5 . It is worth noting that we use the Zero-Shot CoT method to test the basemodel in the GSM8K task, and the results under different Few-Shot settings are also shown in Table 5.

Table 5: Evalution on LoRA-Mixer parameter transferability from Mistral-7B to LLaMA3-8B. Values show absolute performance (relative to baseline in parentheses).

| Method | GSM8K | | | ARC-E | ARC-C |
|---|---|---|---|---|---|
| | 0-shot | 2-shot | 5-shot | 0-shot | 0-shot |
| LLaMA3-8B | 57.92 (1.00) | 75.88 (1.00) | 78.64 (1.00) | **88.45** (1.00) | 78.65 (1.00) |
| + Mistral | **59.13** (1.02) | **76.26** (1.01) | **81.43** (1.04) | 85.89 (0.97) | **79.14** (1.01) |

Table 6: OOD comparison across datasets for Phatgoose, and LoRA-Mixer.

| Dataset | Base | Phatgoose | LoRA-Mixer |
|---------|------|-----------|------------|
| QQP | 68.84 | 69.28 | 69.47 |
| RTE | 70.88 | 69.87 | 71.31 |
| MRPC | 74.73 | 74.59 | 74.93 |

Table 7: Results on BoolQ, HellaSwag, and PIQA. Base is LLaMA3-8B.

| Dataset | Base | LoRA | LoRA-Mixer | Gap |
|---------|------|------|------------|-----|
| BoolQ | 71.25 | 74.46 | 79.37 | +4.91 |
| HellaSwag | 75.33 | 77.39 | 82.41 | +5.02 |
| PIQA | 78.47 | 80.71 | 84.94 | +4.23 |

Interestingly, we observe that we outperform the LLaMA3-8B on two of the three tasks. This cross-model transfer validates the design motivation of LoRA-Mixer and demonstrates that the routing learned via RSL is extremely robust and transferable, making it possible to share experts between models with the same architecture.

To further demonstrate the performance of LoRA-Mixer, we conducted experiments on BoolQ, HellaSwag, and PIQA using LLaMA3-8B, with $r = 32$ for LoRA. The results are shown in Table 7. It can be seen that LoRA-Mixer can effectively coordinate the capabilities of LoRAs to enhance the basemodel. To test the generalization of LoRA-Mixer, we conducted experiments with PHATGOOSE on 9 datasets, six of which were in-distribution experiments and three were out-of-distribution experiments. Here we only show the OOD experimental results because OOD tasks can reflect the generalization ability of LoRA-Mixer. For the results of the in-distribution experiment, please refer to A.14. The results show that LoRA-Mixer has excellent generalization ability, which means that through RSL, routing can make professional decisions and achieve better performance.

### 4.3 Testing on LoRAs Sourced from Internet

To verify that LoRA-Mixer can reuse LoRAs with minimal effort, we tested it on LoRAs from the internet. We downloaded five different LoRAs from LoRAHub Huang et al. (2023) and trained them on SST2, CoLA, MRPC, RTE, and QQP (see Appendix A.15 for details). We used the Flan-T5 model as the base model, froze the LoRAs parameters, and collected an additional 2k mixed data points for routing training, which is independent of the LoRAs training data. The results are shown in Table 3. LoRA-Mixer achieved excellent performance on all four tasks, demonstrating its potential for production-grade multi-task applications.

### 4.4 Comparison with other optimized routing losses.

Although we have compared RSL with other optimized routing losses (MoLE, MixLoRA), to further demonstrate the effectiveness of RSL, we choose three strong baselines for comparison: GMoE, Ds-MoE, and AESL, all of which are specifically optimized for routing losses. It is worth noting that all experiments are conducted with the same training data (2k), and the only difference is the routing loss. As shown in Table 8, RSL significantly outperforms other strong baselines in low-data-resource scenarios.

Table 8: Comparison with GMoE, DS-MoE, and AESL under the same training data (2k) and LoRAs parameters. Demonstrates the advantages of RSL under low data resources.

| Task | GMoE | DS-MoE | AESL | RSL |
|------|------|--------|------|-----|
| SST-2 | 91.38 | 92.45 | 92.64 | **95.41** |
| CoLA | 79.57 | 79.83 | 80.42 | **82.22** |
| ARC-E | 85.65 | 85.32 | 86.24 | **89.88** |
| ARC-C | 76.42 | 78.45 | 79.88 | **83.24** |
| HumanEval | 46.37 | 48.92 | 50.46 | **57.32** |

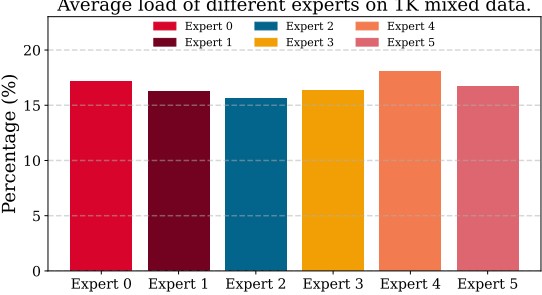

Figure 3: Expert Assignment Overview.

### 4.5 Ablation Study

**Impact of LoRA Rank.** To evaluate the impact of $r$ in LoRAs, we conducted experiments on $r = 16$, $r = 32$, $r = 64$, and $r = 128$, while keeping all other hyperparameters (e.g., dropout rate, learning rate) unchanged. The results of $r = 64$ can be found in Table 2. We place the remaining results in Appendix A.12.

**Expert Load Analysis.** To analyze the overall load, we uniformly sampled 1K data from seven benchmarks (MedicalQA, CoLA, SST2, GSM8K, ARC-E, ARC-C, and Humaneval). We report the average load per expert on this 1K data, as shown in Figure 3 . The activation rates of different experts are relatively balanced, ranging from 15% to 18%. However, across tasks, expert load reflects a "perception" ability, with experts on certain tasks having higher loads than others, as shown in Figure 4 . This demonstrates that our routing mechanism effectively avoids expert collapse and achieves balanced expert utilization across different tasks.

**The impact of K and Cross-domain QA.** We also analyzed the impact of K in Top-K. In general, increasing K within an appropriate range can improve model performance. For details, please refer to A.3. Furthermore, to evaluate LoRA-Mixer's generalization capabilities on cross-domain tasks, we created two cross-domain datasets: Mathematics-Coding and Medical-Mathematics. For detailed experimental results, please refer to A.3. These results demonstrate that LoRA-Mixer maintains strong generalization and input-awareness capabilities on cross-domain datasets.

**The impact of the RSL.** We investigated the impact of our proposed RSL on the LoRA-Mixer. RSL offers three key advantages. First, it enables routing to achieve global load balancing while maintaining robust input awareness. Second, compared to auxiliary losses, RSL requires less training data. Finally, routing trained with RSL is completely independent of the original LoRAs training data.

To verify the first conclusion, we conducted experiments on Medical, GSM8K, and HumanEval. As shown in Figure 4, when using RSL, the router consistently assigns higher activation weights to relevant experts, demonstrating its strong domain awareness and adaptive specialization capabilities. In contrast, when using only the auxiliary loss, the router disregards the semantics of the input and evenly distributes experts, resulting in suboptimal performance.

Table 9: Average performance across seven tasks under different routing training data sizes, with or without RSL. With RSL, LoRA-Mixer requires much less data while showing better performances.

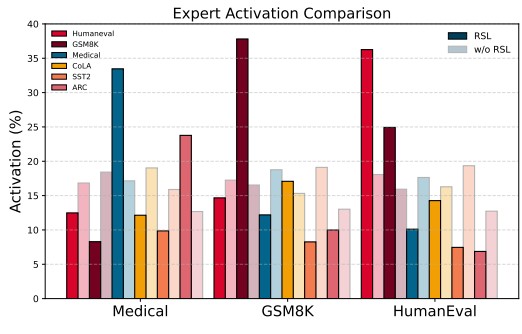

| Data | w/ RSL | w/o RSL | Gap |
|------|--------|---------|------|
| 1K | 76.80 | 75.47 | +1.33 |
| 2K | **79.26** | 77.29 | +1.97 |
| 4K | 78.77 | **79.14** | -0.37 |
| 6K | 79.41 | 79.37 | -0.04 |
| 8K | 79.75 | 79.48 | +0.27 |
| 10K | 79.94 | 79.51 | +0.43 |

Figure 4: Expert Load Distribution across Tasks.

To support the second conclusion, we analyze the impact of training data size on routing performance. Specifically, we construct training sets of different sizes by sampling from a multi-task dataset pool and evaluate the performance on seven benchmark tasks. For clarity, we report the average performance over all tasks.

As shown in Table 9, RSL achieves comparable or even superior performance using only 51.62% of the training data with auxiliary loss. We explain the suboptimal RSL results at 4k in A.16. This exceptionally high data efficiency is demonstrated by generalization bound estimation. See the Appendix A.2 for a detailed proof.

## 5 CONCLUSION.

This paper introduces RSL, designed to address the issue of excessively uniform auxiliary loss. It achieves highly selective routing while achieving overall load balancing, and enables efficient routing training with minimal data, providing strong support for the reuse of LoRAs modules. To verify the effectiveness of RSL, we propose the LoRA-Mixer framework and integrate RSL. LoRA-Mixer is a flexible and architecture-agnostic MoE framework for combining LoRAs and adapting LLMs to multitasking. Unlike other methods that indiscriminately insert MoEs or completely replace attention or FFN modules, LoRA-Mixer only adapts the core projection layer, improving the performance of Transformers and SSM models. Although LoRA-Mixer is effective, its fixed top-$K$ routing may limit adaptability to ambiguous inputs. Uniformly applying it across all layers can also introduce redundancy, as different layers capture different information. Future work will explore dynamic or differentiable routing and adaptive integration to apply LoRA-Mixer only where most beneficial.

## 6 ACKNOWLEDGMENTS

This research was supported by the National Natural Science Foundation of China (Nos. 62272184 and 62402189), the China Postdoctoral Science Foundation (Nos. 2024M751012, 2025T180429, and GZC20230894), and the Hubei Provincial Postdoctoral Research Funding Program (No. 2024HBB-HCXB014). The computational work was performed on the high-performance computing platform at Huazhong University of Science and Technology.

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

# A APPENDIX

## A.1 OPTIMIZATION AND CONVERGENCE ANALYSIS.

We analyze the empirical counterpart of $\mathcal{L}_{\text{RSL}}$ on a dataset $S = \{x_j\}_{j=1}^n$:

$$\widehat{\mathcal{L}}_{\text{RSL}}(S; \{p(x_j)\}) = \alpha \sum_{i=1}^K \widehat{p}_i \cdot \widehat{\overline{f}}_i - \frac{\lambda}{n} \sum_{j=1}^n \mathcal{H}(p(x_j)),$$

$$\widehat{p}_i = \frac{1}{n} \sum_{j=1}^n p_i(x_j), \qquad \widehat{\overline{f}}_i = \frac{1}{n} \sum_{j=1}^n \mathbb{I}(i = \arg\max_\ell p_\ell(x_j)). \tag{13}$$

**Smooth surrogate for optimization.** For optimization analysis we replace the hard top-1 frequency by a standard smooth surrogate:

**Assumption 1** (Smoothed usage surrogate). *Replace $\widehat{\overline{f}}_i$ in equation 13 by $\widehat{\overline{s}}_i$, where $s_i^\tau(x) = \mathrm{softmax}(G(x)/\tau)_i$ for some $\tau > 0$ and $\widehat{\overline{s}}_i = \frac{1}{n} \sum_j s_i^\tau(x_j)$. We assume the composite $\sum_i \widehat{p}_i \widehat{\overline{s}}_i$ is convex and $L$-smooth in $\{p(x_j)\}$ on the product simplex $\Delta^n$.[1]*

**Negative-entropy–regularized objective.** Define

$$F_S(\{p(x_j)\}) := \alpha \sum_{i=1}^K \widehat{p}_i \cdot \widehat{\overline{s}}_i + \frac{\lambda}{n} \sum_{j=1}^n \sum_{i=1}^K p_i(x_j) \log p_i(x_j). \tag{14}$$

Since $-\sum_i p_i \log p_i = \mathcal{H}(p)$, minimizing $F_S$ is identical to minimizing $\widehat{\mathcal{L}}_{\text{RSL}}$ with $\widehat{\overline{f}}_i$ replaced by $\widehat{\overline{s}}_i$; a change of logarithm base is absorbed into $\lambda$. We optimize $F_S$ over $\{p(x_j)\} \in \Delta^n$ using entropic mirror descent (exponentiated gradient)

$$p(x_j)^{t+1} \propto p(x_j)^t \odot \exp\Big(-\eta_t \nabla_{p(x_j)} F_S(\{p(x_\ell)^t\})\Big), \quad \text{then normalize onto } \Delta. \tag{15}$$

**Lemma 1** (Strong convexity). *The function $h(p) = \sum_{i=1}^K p_i \log p_i$ is 1-strongly convex on $\Delta$ w.r.t. the $\ell_1$ norm. Hence the regularizer $\frac{\lambda}{n} \sum_{j=1}^n h(p(x_j))$ is $\lambda$-strongly convex on the product simplex $\Delta^n$. Therefore, $F_S$ in equation 14 is $\lambda$-strongly convex, since the usage surrogate term is convex by Assumption 1.*

**Lemma 2** (Gradient bounds and interior smoothness). *Under Assumption 1 there exists $G > 0$ such that for all feasible $\{p(x_j)\}$,*

$$\Big\| \nabla_{p(x_j)} \Big( \alpha \sum_{i=1}^K \widehat{p}_i \cdot \widehat{\overline{s}}_i \Big) \Big\| \le G.$$

*Moreover,*

$$\Big\| \nabla_{p(x_j)} \frac{\lambda}{n} \sum_{i=1}^K p_i(x_j) \log p_i(x_j) \Big\| \le \frac{\lambda}{n} \big\| 1 + \log p(x_j) \big\|,$$

*so if we maintain $p_i(x_j) \ge \epsilon$ during training (e.g., via temperature/clipping), the right-hand side is uniformly bounded by a constant $C(\lambda, \epsilon, K)$. On the interior $\{p : p_i \ge \epsilon\}$, the entropy term is $L_{ent} \le \frac{\lambda}{n\epsilon}$-smooth (w.r.t. $\ell_2$), hence $F_S$ is $L$-smooth with $L \le L_{ent} + L_{aux}$ for some finite $L_{aux} = O(\alpha/n)$.*

**Theorem 1** (Convergence of EG/EMD for $F_S$). *Under Assumption 1 and Lemmas 1–2, $F_S$ is $\lambda$-strongly convex on $\Delta^n$. (a) Sublinear rate. With steps $\eta_t = \frac{1}{\lambda t}$, the averaged iterate $\bar{P}^T = \frac{1}{T} \sum_{t=1}^T P^t$ satisfies*

$$F_S(\bar{P}^T) - F_S(P^\star) \le \mathcal{O}\Big( \frac{G_{\text{eff}}^2}{\lambda T} \Big),$$

---

[1] This includes the commonly used choice $\widehat{\overline{s}}_i = \widehat{p}_i$, giving the auxiliary term $\alpha \sum_i \widehat{p}_i^2$, which is convex since each $\widehat{p}_i$ is linear in $\{p(x_j)\}$.

*where $G_{\text{eff}}$ aggregates the uniform gradient bounds in Lemma 2.* ***(b) Linear rate on the interior.*** *If, in addition, training maintains $p_i(x_j) \geq \epsilon$ (hence $F_S$ is L-smooth on the interior), then with any constant step $\eta \leq 1/L$ the EG iterates enjoy geometric contraction in KL divergence:*

$$D_{\text{KL}}(P^\star \parallel P^{t+1}) \leq (1 - \eta\lambda) \, D_{\text{KL}}(P^\star \parallel P^t),$$

*and consequently $F_S(P^t) - F_S(P^\star)$ also decays geometrically.*

The negative-entropy term supplies curvature in the routing simplex (Lemma 1), turning the merely convex surrogate into a *strongly convex* objective. This stabilizes and accelerates routing optimization while preserving input-awareness through the token-level $\log p_i(x)$ signals (cf. the gradient form in the main RSL, Eq. equation 9).

## A.2 GENERALIZATION BOUND VIA ALGORITHMIC STABILITY.

We analyze generalization under the *same* smoothed usage surrogate as in optimization (Assumption 1), i.e., replacing $\widehat{f}_i$ by $\widehat{s}_i$ (in particular, the commonly used choice $\widehat{s}_i = \widehat{p}_i$ yields a convex quadratic auxiliary term). We also keep the empirical objective in *sample-average* form so that the regularization strength does not scale with $n$.

Let $S = \{x_j\}_{j=1}^n$ and $S^{(i)}$ be $S$ with the $i$-th example replaced. Denote by $\hat{P}(S)$ a (unique) minimizer of the empirical objective

$$F_S(\{p(x_j)\}) = \alpha \sum_{i=1}^{K} \widehat{p}_i \cdot \widehat{s}_i + \frac{\lambda}{n} \sum_{j=1}^{n} \sum_{i=1}^{K} p_i(x_j) \log p_i(x_j),$$

and similarly $\hat{P}(S^{(i)})$ for $F_{S^{(i)}}$. For reference, the population counterpart is

$$F(\{p(x)\}) = \alpha \sum_{i=1}^{K} \bar{p}_i \cdot \bar{s}_i + \lambda \, \mathbb{E}_x\Big[ \sum_{i=1}^{K} p_i(x) \log p_i(x) \Big], \quad \bar{p}_i = \mathbb{E}_x[p_i(x)], \ \ \bar{s}_i = \mathbb{E}_x[s_i^\tau(x)].$$

**Assumption 2** (Lipschitz loss & bounded differences). *(i) The map $\{p(x_j)\} \mapsto \alpha \sum_{i=1}^{K} \widehat{p}_i \cdot \widehat{s}_i$ is L-Lipschitz on the product simplex endowed with the averaged block norm $\|P\|_{\text{avg-1}} := \frac{1}{n} \sum_j \|p(x_j)\|_1$, and replacing one sample in $S$ changes this auxiliary term by at most $\frac{B}{n}$ (bounded-difference property). (ii) The per-example evaluation loss $\ell(p; x)$ is $\rho$-Lipschitz in $p$ (under $\|\cdot\|_1$ on a single simplex) and bounded in $[0, M]$.*

**Lemma 3** (Strong convexity under the averaged product norm). *Negative entropy $h(p) = \sum_{i=1}^{K} p_i \log p_i$ is 1-strongly convex on $\Delta$ w.r.t. $\|\cdot\|_1$. Therefore $\frac{\lambda}{n} \sum_{j=1}^{n} h(p(x_j))$ is $\lambda$-strongly convex on the product simplex endowed with $\|\cdot\|_{\text{avg-1}}$. Since the surrogate usage term is convex (Assumption 1), $F_S$ is $\lambda$-strongly convex.*

**Lemma 4** (Uniform stability of ERM for $F_S$). *Under Assumptions 1 and 2, let $\hat{P}(S)$ and $\hat{P}(S^{(i)})$ be the ERM solutions. Then the solution mapping is uniformly stable:*

$$\big\| \hat{P}(S) - \hat{P}(S^{(i)}) \big\|_{\text{avg-1}} \leq \mathcal{O}\Big( \frac{L+B}{\lambda \, n} \Big).$$

**Theorem 2** (Generalization bound for RSL-trained router). *Under Assumptions 1 and 2, the expected generalization gap satisfies*

$$\big| \, \mathbb{E}_x[\ell(\hat{P}(S); x)] - \frac{1}{n} \sum_{j=1}^{n} \ell(\hat{P}(S); x_j) \, \big| = \mathcal{O}\Big( \frac{\rho \, (L+B)}{\lambda \, n} \Big),$$

*and, with probability at least $1 - \delta$,*

$$\big| \, \mathbb{E}_x[\ell(\hat{P}(S); x)] - \frac{1}{n} \sum_{j=1}^{n} \ell(\hat{P}(S); x_j) \, \big| \leq \mathcal{O}\Big( \frac{\rho \, (L+B)}{\lambda \, n} \Big) + \mathcal{O}\Big( \sqrt{\frac{\log(1/\delta)}{n}} \Big).$$

Lemma 3 gives $\lambda$-strong convexity of $F_S$ under $\|\cdot\|_{\text{avg-1}}$. By a standard argument for strongly convex ERM with bounded differences, replacing one sample perturbs the objective by at most $O((L+B)/n)$ and yields uniform stability $\beta = \mathcal{O}((L+B)/(\lambda n))$ for the solution map (Lemma 4). The Lipschitz evaluation loss then converts stability into generalization: $\mathbb{E}[\text{gap}] \leq \rho\beta$, and a standard McDiarmid/Hoeffding argument implies the high-probability term $\mathcal{O}(\sqrt{\log(1/\delta)/n})$.

This matches the empirical robustness of RSL in low-data regimes: entropy regularization both improves optimization conditioning and reduces the hypothesis sensitivity to single-sample perturbations.

### A.3 ADDITIONAL EXPERIMENTAL RESULTS.

**The impact of Top-$K$.** To explore the impact of Top-K routing, we conducted experiments on SST-2 and CoLA using Falcon-Mamba. The results are shown in Figure 5. As $K$ increases from 1 to 3, we observe improved accuracy, suggesting that multiple experts can gain complementary information. However, further increasing $K$ may actually degrade performance. Therefore, how to set or dynamically learn the optimal $K$ value is a direction worthy of further research.

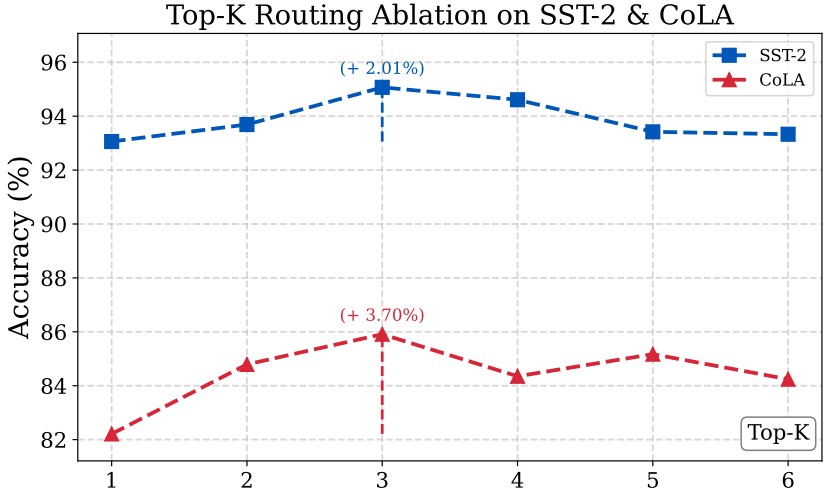

Figure 5: Top-K Routing Impact.

**Cross-domain QA** To evaluate LoRA-Mixer's cross-domain generalization capabilities, we constructed two question-answering datasets: Medical-Mathematics and Mathematics-Coding. Each dataset contains 200 examples generated by DeepSeek-R1. These questions are more challenging. We conduct DeepSeek-R1 evaluations, and the results are shown in Table 10.

Table 10: Cross-domain performance of LoRA-Mixer on LLaMA3-8B.

| Task | Base | LoRAHub | MOLE | MixLoRA | LoRA-Mixer |
|------|------|---------|------|---------|------------|
| Math-Medical | 69.88 | 70.53 | 72.11 | 72.74 | **73.41** |
| Math-Coding | 59.37 | 61.08 | 62.24 | 63.10 | **63.46** |

### A.4 PARAMETER ANALYSIS.

We tested the trainable parameters of LoRA-Mixer, MoLE, MixLoRA, and LoRAHub under LLaMA3-8B. The trainable parameters of LoRA-Mixer and MixLoRA are 3.88% and 8.08%, respectively (six experts + routing). LoRA-Mixer's total parameters account for 48% of MixLoRA's, with routing parameters accounting for 0.04%. MoLE's training parameters are routing parameters. To ensure a fair comparison, we adjusted MoLE's routing to reduce the trainable parameters to 0.04%. LoRAHub is a train-free method with 0% trainable parameters. PHATGOOSE's trainable parameters are gating vectors, specifically 0.01%.

### A.5 DATASETS DETAILS.

We present information about all datasets used here for reference.

Table 11: Statistics of datasets used in experiments.

| Dataset | Train | Dev | Test | Task |
|---------|-------|-----|------|------|
| ARC-E | 2251 | 570 | 2376 | Multiple-choice QA |
| ARC-C | 1119 | 299 | 1172 | Multiple-choice QA |
| GSM8K | 7473 | – | 1319 | Math reasoning |
| MBPP | 779 | – | 195 | Code generation |
| CoLA | 8551 | 1043 | 1063 | Acceptability judgment |
| SST-2 | 10000 | 872 | 1821 | Sentiment classification |
| Medical | 10030 | 4004 | 4004 | Medical QA |
| MRPC | 3668 | 408 | 1725 | Paraphrase detection |
| MNLI | 392702 | 9815 | 9832 | Natural language inference |
| QNLI | 104743 | 5463 | 5463 | Question answering/NLI |
| RTE | 2490 | 277 | 3000 | Natural language inference |
| QQP | 363846 | 40430 | 390965 | Paraphrase identification |
| PIQA | 16113 | 1838 | 3277 | Physical commonsense reasoning |
| HellaSwag | 39905 | 10042 | 10042 | Commonsense reasoning |
| BoolQ | 9427 | 3270 | 3270 | Boolean question answering |

### A.6 COMPUTATIONAL RESOURCES AND REPRODUCIBILITY GUARANTEE.

Our experiments were conducted on a Linux workstation equipped with a single NVIDIA A800 80GB GPU and a 32-core Intel Xeon CPU, using PyTorch version 2.1.0 and Transformers version 4.36.2. For the first stage of training, we used LoRA with parameters r = 64, alpha = 128, dropout = 0.1, batch size = 1, gradient accumulation = 4, and lr = 1e-5. The task loss function was cross-entropy loss, and a cosine scheduler with warmup ratio (ratio = 0.1) was used. In stage 2, RSL was added with $\alpha = 0.1$ and $\lambda = 0.5$ in RSL. $\beta$ of $\mathcal{L}_{\text{preserve}}$ was fixed to 5e-5, and a top-3 routing strategy was used. In the inference phase, we also adopt the top-3 routing strategy and test it in the same device environment.

### A.7 EFFICIENCY AND RESOURCE OVERHEAD ANALYSIS.

We tested inference/training time, and memory usage, and the results are shown in the table 12. These metrics highlight the advantages of LoRA-Mixer beyond task performance. Inference was performed using a single A800 80G GPU, with max_length=512 and max_new_tokens=15, to test single-sample inference time. To eliminate the effects of cold starts, all measurements were performed after a warm-up run.

Table 12: Inference time comparison across different methods.

| Model | Time (s) |
|-------|----------|
| LLaMA3-8B (Baseline) | 0.441 |
| LoRAHub | 0.482 |
| MoLE | 0.563 |
| MixLoRA | 0.597 |
| LoRA-Mixer (Ours) | 0.574 |

LoRAHub uses a training-free approach, resulting in faster inference speed. MoLE is nearly identical to LoRA-Mixer. However, this coarse-grained approach often leads to poor performance. Overall, LoRA-Mixer achieves a better balance between inference speed and performance. Overall, LoRA-Mixer optimized with RSL achieves faster convergence and more stable training. For training information, LoRA-Mixer's training is split into two stages: Stage 1 (expert training) and Stage 2 (router training). The results are shown in the table 13.

Table 13: Training stages with dataset size, time.

| Stage | Data | Time (hours) |
|---|---|---|
| Stage 1 | 40k | 8.86 |
| Stage 2 | 2k | 1.74 |
| Stage 2 | 4k | 3.38 |

LoRA-Mixer can skip stage 1 and quickly complete the stage 2 of training using very little data when computing resources are limited, and provide strong performance. This is very important in resource-constrained scenarios. For memory usage, we measure memory usage for 6 experts, r=64 (A800-80G, float32, batchsize=1, gradientaccumulation=4),the results are shown in the table 14:

Table 14: Memory usage comparison across different methods.

| Method | MixLoRA | MoLE | LoRA-Mixer |
|---|---|---|---|
| Memory Usage (GB) | 43.07 | 36.57 | 38.48 |

## A.8 HYPERPARAMETER GRID SEARCH.

The hyperparameters include $\alpha$,$\beta$ and $\lambda$ are not many, and we design a simple yet effective strategy for tuning the hyperparameters. We start with heuristic values informed by prior work on multi-objective optimization: $\alpha = 0.1$ and $\lambda = 0.05$ in $\mathcal{L}_{RSL}$. We fix $\beta = 5e - 5$ in $\mathcal{L}_{preserve}$. These values are chosen to ensure that the task loss dominates the training in the early stages of training. We perform grid search over a narrow range of $\alpha$ and $\lambda$ values on a validation split of all training data (10% of the total data).

Table 15: Ablation study on $\alpha$ and $\lambda$. Best results are highlighted in bold.

| $\alpha$ | $\lambda$ | Performance | Load Balance (Variance) | Routing Entropy |
|---|---|---|---|---|
| 0.05 | 0.01 | 76.2% | 0.12 | 1.9 |
| **0.1** | **0.05** | **78.5%** | **0.08** | **1.5** |
| 0.2 | 0.1 | 75.8% | 0.05 | 1.2 |

The values $\alpha = 0.1$ and $\lambda = 0.05$ achieve the best trade-off across all metrics. To prevent RSL from interfering too much with task loss in the early stages of training, we dynamically adjust parameters during training, for example, gradually increasing $\alpha$ and $\lambda$ to their final values in the first 30% of epochs and then keeping them constant. This "warm-up" strategy stabilizes training and improves convergence.

## A.9 ADDITIONAL RESULTS ON GLUE TASKS.

We present the results of LoRA-Mixer on the QNLI and MNLI tasks, as shown in Table 16. The results show that LoRA-Mixer achieves state-of-the-art results on all GLUE tasks, demonstrating the powerful combination of RSL and LoRA-Mixer. The experimental settings are LLaMA3-8B and $r = 16$, and the average performance over three runs is reported.

Table 16: Results on QNLI and MNLI. Higher is better.

| Method | QNLI | MNLI |
|---|---|---|
| Base | 90.36 | 89.95 |
| LoRAHub | 90.44 | 90.56 |
| MOLE | 91.48 | 91.04 |
| MixLoRA | 91.73 | 91.27 |
| LoRA | 91.34 | 90.87 |
| **LoRA-Mixer (Ours)** | **92.04** | **91.45** |

We exclude STSB because it focuses on measuring semantic similarity, which is inconsistent with our core goal of "cross-task expert collaboration" (classification, reasoning, and generation), and the results are more likely to be random. Furthermore, we want to be consistent with previous studies such as GOAT, LoRA-LEGO, and MixLoRA, which did not conduct experiments on this dataset.

### A.10  ADDITIONAL EMPIRICAL SUPPORT FOR LoRA MIGRATION.

To better understand LoRA-Mixer's cross-model transfer experiments, we analyzed the architectures of Mistral-7B and LLaMA3-8B. The Mistral-7B and LLaMA3-8B models share a core decoder-specific Transformer structure, and the key dimensions match, as shown below:

Table 17: Comparison of architecture components between Mistral-7B and LLaMA3-8B.

| Architecture components | Mistral-7B | LLaMA3-8B |
|---|---|---|
| Number of layers | 32 | 32 |
| Hidden dimension ($d_{\text{model}}$) | 4096 | 4096 |
| Attention heads | 32 | 32 |
| Feedforward dimension ($d_{\text{ff}}$) | 14336 | 14336 |
| Attention projection layer | Q, K, V, output (all $4096 \rightarrow 4096$) | Q, K, V, output (all $4096 \rightarrow 4096$) |
| Activation function | SwiGLU | SwiGLU |

This structural alignment ensures that linear projection layers with the same input/output dimensions can achieve parameter transfer.

### A.11  FALCON-MAMBA ARCHITECTURE ANALYSIS.

Mamba builds upon the state space model. It processes an input sequence $x(t) \in \mathbb{R}^L$ to produce an output $y(t) \in \mathbb{R}^L$ by employing a hidden state $h(t) \in \mathbb{R}^N$. This relationship is initially defined by a continuous system:

$$\begin{aligned} h'(t) &= \mathbf{A}h(t) + \mathbf{B}x(t), \\ y(t) &= \mathbf{C}h(t). \end{aligned} \tag{16}$$

Here, $\mathbf{A} \in \mathbb{R}^{N \times N}$ is the state transition matrix, and $\mathbf{B} \in \mathbb{R}^{N \times 1}$, $\mathbf{C} \in \mathbb{R}^{N \times 1}$ are projection matrices.

To process discrete sequences, Mamba discretizes the continuous parameters $\mathbf{A}$ and $\mathbf{B}$ using a time scale parameter $\Delta$ and the zero-order hold (ZOH) principle, resulting in discretized parameters $\overline{\mathbf{A}}$ and $\overline{\mathbf{B}}$:

$$\begin{aligned} \overline{\mathbf{A}} &= \exp\left(\mathbf{\Delta A}\right), \\ \overline{\mathbf{B}} &= \left(\mathbf{\Delta A}\right)^{-1}\left(\exp\left(\mathbf{\Delta A}\right) - \mathbf{I}\right) \cdot \mathbf{\Delta B}. \end{aligned} \tag{17}$$

The discrete state-space equation with a step size of $\Delta$ becomes:

$$\begin{aligned} h_t &= \overline{\mathbf{A}}h_{t-1} + \overline{\mathbf{B}}x_t, \\ y_t &= \mathbf{C}h_t. \end{aligned} \tag{18}$$

By iteratively expanding the hidden state $h_{t-1}$, Mamba derives a global convolution kernel $\overline{\mathbf{K}} \in \mathbb{R}^L$. This kernel is then used to compute the output $y$ through a convolution operation with the input $x$:

$$\overline{\mathbf{K}} = \left( \mathbf{C}\overline{\mathbf{B}}, \mathbf{C}\overline{\mathbf{A}}\overline{\mathbf{B}}, ..., \mathbf{C}\overline{\mathbf{A}}^{L-1}\overline{\mathbf{B}} \right),$$

$$y = x \otimes \overline{\mathbf{K}}. \tag{19}$$

Falcon Mamba 7B adopts a pure Mamba architecture, a departure from hybrid designs incorporating staggered attention. This deliberate choice aims to maintain the intrinsic linear scalability characteristic of Mamba models. To enhance adaptability, the model employs decoupled input embeddings and output weights.

At its core, Falcon-Mamba features 64 layers of the Falcon-Mamba Mixer. Each Mixer layer integrates an SSM (State Space Model) module alongside in-projection and out-projection layers, RMS Norm, and a convolutional layer.

Within the SSM module, the input is mapped to $\Delta$, $B$, and $C$ through a projection layer denoted as $x$-proj:

$$x \xrightarrow{x\text{-proj}} (\Delta, B, C)$$

where $x$ represents the input to the SSM module. Furthermore, another projection layer, $dt$-proj, discretizes $\Delta$:

$$\Delta \xrightarrow{dt\text{-proj}} \Delta_{discretized}$$

These discretized values—$\Delta_{discretized}$, $A$, $B$, $C$, and $D$—are then fed into the selective scanning module for processing. This architectural design of Falcon-Mamba fully enables the application of LoRA-Mixer specifically tailored for the projection layer.

## A.12 THE IMPACT OF $r$ IN LORAS

Table 18: Comparison of LoRA-Mixer on Falcon-Mamba, Mistral, and LLaMA across seven tasks. LoRA denotes single-task fine-tuning with rank $r = 16$. Best results per model and task are highlighted in bold.

| Methods | Medical | CoLA | SST2 | GSM8K | ARC-E | ARC-C | HumanEval |
|---|---|---|---|---|---|---|---|
| FalconMamba-LoRA | 76.33 | 82.75 | 93.23 | 54.62 | 83.97 | 76.08 | 28.66 |
| +LoRA-Mixer | **77.03** | **83.80** | **93.41** | **55.15** | **84.17** | **76.51** | **29.48** |
| Mistral-LoRA | 67.87 | 75.55 | 89.14 | **45.96** | 84.37 | 69.51 | **34.76** |
| +LoRA-Mixer | **68.27** | **77.64** | **90.27** | 45.61 | **84.46** | **70.15** | 34.68 |
| LLaMA-LoRA | 79.35 | 77.65 | 94.15 | 61.79 | 88.64 | 79.47 | 51.78 |
| +LoRA-Mixer | **79.88** | **78.11** | **94.97** | **61.14** | **89.29** | **79.87** | **53.39** |

Table 19: Comparison of LoRA-Mixer on Falcon-Mamba, Mistral, and LLaMA across seven tasks. LoRA denotes single-task fine-tuning with rank $r = 32$. Best results per model and task are highlighted in bold.

| Methods | Medical | CoLA | SST2 | GSM8K | ARC-E | ARC-C | HumanEval |
|---|---|---|---|---|---|---|---|
| Falcon-Mamba-LoRA | 76.32 | 85.90 | 93.12 | 54.76 | 84.86 | 75.67 | 32.33 |
| +LoRA-Mixer | **76.67** | **86.00** | **95.35** | **55.41** | **85.55** | **76.81** | **34.15** |
| Mistral-LoRA | 68.57 | 76.89 | 93.87 | **46.29** | 84.87 | 68.83 | 32.93 |
| +LoRA-Mixer | **68.88** | **78.81** | **94.60** | 45.91 | **85.91** | **71.80** | **33.56** |
| LLaMA-LoRA | 79.15 | 81.11 | 95.30 | 61.38 | 88.76 | 79.31 | 52.34 |
| +LoRA-Mixer | **80.87** | **81.30** | **95.53** | **62.46** | **89.04** | **79.48** | **53.65** |

Table 20: Comparison of LoRA-Mixer on Falcon-Mamba, Mistral, and LLaMA across seven tasks. LoRA denotes single-task fine-tuning with rank $r = 128$. Best results per model and task are highlighted in bold.

| Methods | Medical | CoLA | SST2 | GSM8K | ARC-E | ARC-C | HumanEval |
|---|---|---|---|---|---|---|---|
| Falcon-Mamba-LoRA | 76.42 | 85.25 | 92.88 | 55.25 | 83.91 | 75.72 | 32.41 |
| +LoRA-Mixer | **76.81** | **85.97** | **94.30** | **56.11** | **85.50** | **77.75** | **33.10** |
| Mistral-LoRA | 69.63 | 79.85 | 90.14 | **46.05** | 84.62 | 68.59 | 34.87 |
| +LoRA-Mixer | **69.82** | **80.75** | **91.55** | 44.55 | **85.85** | **71.74** | **35.17** |
| LLaMA-LoRA | 79.38 | 81.48 | 95.27 | 62.04 | 89.21 | 80.28 | 55.40 |
| +LoRA-Mixer | **80.82** | **82.25** | **95.50** | **63.41** | **89.33** | **81.43** | **56.31** |

As shown in Table 18,Table 19 and Table 20, within a certain range, as $r$ increases, the performance of the model can be improved to a certain extent. Our method is not only better than the basic model, but even better than the fine-tuned basic model on most tasks. This shows that our method can effectively combine existing knowledge through dynamic expert combination to form a more "intelligent" model.

## A.13 BALANCE LOSS VISUALIZATION.

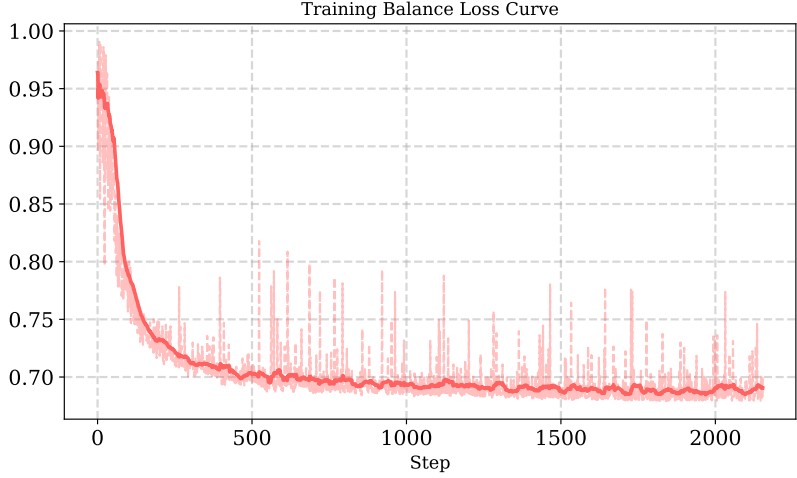

Figure 6: Balance loss curve using RSL loss during training.

Figure 6 shows the changing trend of the Balance Loss when the RSL loss function is used during training. As shown in the figure, the Balance Loss drops rapidly in the early stage of training, indicating that our model can quickly learn an effective expert routing strategy, thanks to the synergy of the global consistency term and the local entropy penalty term in the RSL loss function. In the middle stage of training, the Balance Loss continues to drop steadily with a small fluctuation, which reflects the stability that the RSL loss function brings to the training process. In the late stage of training, the Balance Loss remains stable at a low level, further demonstrating the balance and optimization effect of the model in the use of experts. Overall, this Balance Loss curve not only reflects the model's ability to converge quickly, but also demonstrates the robustness of the training process, verifying the significant advantages of the RSL loss function in improving model performance and training efficiency.

## A.14   IN-DISTRIBUTION EXPERIMENTAL RESULTS.

Here, we present experimental results for LoRA-Mixer and PHATGOOSE on five in-distribution experiments (GSM8K, ARC-E, ARC-C, CoLA, and SST2). The results are shown in Table 21. Both methods use the same LoRA adapter (Q/K/V/O) on LLaMA3-8B and are trained using the same data; their routing strategies differ: LoRA-Mixer uses RSL, while PHATGOOSE uses sigmoid gating based on cosine similarity. It can be seen that LoRA-Mixer outperforms PHATGOOSE across the board.

Table 21: Comparison of Base, Phatgoose, and LoRA-Mixer across five datasets.

| Dataset | Base | Phatgoose | LoRA-Mixer |
|---------|------|-----------|------------|
| SST2 | 93.12 | 91.35 | **94.83** |
| CoLA | 79.14 | 78.27 | **81.17** |
| GSM8K | 57.92 | 58.87 | **62.66** |
| ARC-E | 88.45 | 87.13 | **89.47** |
| ARC-C | 78.65 | 75.94 | **79.89** |

In order to verify the enhanced performances of individual expert after LoRA-Mixer optimization. We selected four tasks, GSM8K, SST2, CoLA and HumanEval, for experiments. The results are shown in Table **??**. We can find that LoRAs optimized by LoRA-mixer exhibit improved performance, especially in the GSM8K task, where the performance improved by **5.36%** after adding mathematical experts. This result confirms the enhanced ability of each individual LoRA expert after LoRA-Mixer optimization.

## A.15   DETAILS OF LoRA MODULES OF FLAN-T5.

To better understand the Flan-T5 LoRAs experiment from the Internet, we introduce the corresponding LoRAs configuration information here, details as 22:

Table 22: LoRA configuration details used in our experiments.

| Parameter | Value |
|-----------|-------|
| base_model_name_or_path | google/flan-t5-large |
| bias | none |
| fan_in_fan_out | false |
| inference_mode | true |
| init_lora_weights | true |
| layers_pattern | null |
| layers_to_transform | null |
| lora_alpha | 32 |
| lora_dropout | 0.1 |
| modules_to_save | null |
| peft_type | LORA |
| r | 16 |
| revision | null |
| target_modules | [q, v] |
| task_type | SEQ_2_SEQ_LM |

## A.16   EXPLANATION OF RSL'S SUBOPTIMAL PERFORMANCE WITH 4K DATA.

RSL's temporary performance fluctuations on 4k datasets can be explained by its design philosophy. RSL integrates an entropy regularizer to balance expert specialization and load balancing. On small datasets (1k-3k instances), this mechanism helps the router quickly focus on the most relevant experts by avoiding overly uniform activation functions, thus outperforming common auxiliary loss functions that tend to enforce an indiscriminate balance. When the number of instances reaches 4k, RSL begins to explore finer-grained expert tasks (consistent with its goal of input-aware routing). However,

this phase can lead to temporary instability, as the data volume is sufficient to trigger exploration but not yet rich enough to fully stabilize finer routing patterns. In contrast, the original auxiliary loss function maintains a stable but suboptimal uniform activation function, showing a temporary advantage. However, as data size increases further, RSL's advantage increases. This is because the auxiliary loss becomes overly averaged and loses input awareness, resulting in less accurate expert selection and suboptimal performance.

## A.17 EXCESSIVE AVERAGING OF AUXILIARY LOSSES.

The standard auxiliary loss in mixture-of-experts is designed to encourage load balancing by aligning the average routing probability $\bar{p}_i$ with the average usage frequency $\bar{f}_i$:

$$L_{\text{aux}} = \alpha \sum_{i=1}^{K} \bar{p}_i \, \bar{f}_i, \qquad \sum_{i=1}^{K} \bar{p}_i = 1. \tag{20}$$

Here $\bar{p}_i = \mathbb{E}[p_i(x)]$ and $\bar{f}_i = \mathbb{E}[\mathbf{1}\{i = \arg\max_j p_j(x)\}]$. Since $\bar{f}_i$ depends on hard top-1 routing and is not differentiable, implementations usually adopt a smooth surrogate $\bar{s}_i$, e.g. softmax usage with temperature $\tau > 0$. With this surrogate, the alignment loss is often combined or replaced with a quadratic balancing form:

$$L_{\text{bal}} = \alpha \sum_{i=1}^{K} \bar{p}_i^2, \qquad \sum_{i=1}^{K} \bar{p}_i = 1, \tag{21}$$

which penalizes deviations from equal usage. Consider the Lagrangian

$$\mathcal{L}(\bar{p}, \lambda) = \sum_{i=1}^{K} \bar{p}_i^2 - \lambda \Big( \sum_{i=1}^{K} \bar{p}_i - 1 \Big). \tag{22}$$

Setting $\partial \mathcal{L}/\partial \bar{p}_i = 0$ yields $2\bar{p}_i - \lambda = 0 \Rightarrow \bar{p}_i = \lambda/2$ for all $i$. Summing over $i$ gives $\lambda = 2/K$, hence

$$\bar{p}_i^\star = \frac{1}{K}, \qquad \forall i. \tag{23}$$

The unique minimizer of $L_{\text{bal}}$ is the uniform distribution, showing that traditional auxiliary losses inherently bias the router towards equal activation and thus hinder expert specialization.

## A.18 USE OF LLMs.

We use LLMs as a basemodel for our experiments, and we also use LLMs for syntax polishing and checking.

