# OpenReview forum: "LoRA-Mixer: Coordinate Modular LoRA Experts Through Serial Attention Routing"
_ICLR.cc/2026/Conference — ICLR 2026 Poster_

### Official Review · Reviewer_EiuV · 2025-10-20

**Soundness:** 2
**Presentation:** 3
**Contribution:** 2
**Rating:** 4
**Confidence:** 4

**Summary:**

This paper introduces LoRA-Mixer, a modular mixture-of-experts (MoE) framework for combining multiple LoRA adapters through serial attention routing at the projection layers of Transformers or state-space models (SSMs). Instead of replacing full attention or FFN blocks, LoRA-Mixer injects multiple LoRA “experts” directly into the linear projections and learns a lightweight router optimized by a novel Routing Specialization Balance Loss (RSL). RSL integrates entropy-based specialization and global load balancing, encouraging both expert diversity and stability. The authors claim that this approach achieves fine-grained token-level routing with substantially fewer trainable parameters. Experiments show consistent gains over LoRA-MoE baselines such as MixLoRA, MoLE, and LoRAHub while using only ~48 % of their parameters.

**Strengths:**

(1) the paper introduces a simple yet effective way to route LoRA experts through attention projections, improving modularity without large architectural changes.

(2) The proposed RSL loss is a reasonable refinement that balances specialization and load, supported by clear theoretical grounding.

(3) Experiments across multiple models and tasks show consistent, reproducible gains with good parameter efficiency.

**Weaknesses:**

(1) The efficiency claim relies mainly on the “48% trainable parameters” figure, without corresponding latency, throughput, FLOPs, or memory comparisons under matched hardware conditions.

(2) The paper does not clarify whether routers are defined per-projection (Q/K/V/O), shared per layer, or applied only to selected layers, which affects both computational cost and specialization behavior.

(3) The influence of RSL coefficients (λ, α), preservation regularizer β, and Top-K settings is not systematically reported; the cited appendices lack quantitative curves or trend analyses.

(4) Figures primarily show average expert load across tasks rather than per-token entropy or variance, leaving the “input-aware routing” claim insufficiently supported.

(5) Transfer from Mistral-7B to LLaMA3-8B improves GSM8K but slightly degrades ARC-E, indicating partial rather than robust transferability.

(6) While several routing-loss baselines are included, recent MoE-LoRA variants from 2024–2025 (e.g., DynMoLE, MORAL) under matched protocols are missing, limiting external comparison.

**Questions:**

(1) Could the authors provide runtime measurements (training/inference latency, FLOPs, or GPU memory) under matched hardware?

(2) Please clarify whether the router operates per-projection (Q/K/V/O), shared per layer, or selectively on certain layers.

(3) Could the authors include quantitative ablations or trend plots showing the sensitivity of RSL coefficients (λ, α), preservation regularizer β, and Top-K to model performance?

(4) Would it be possible to visualize token-level routing entropy or variance to directly support the claim of input-aware specialization?

(5) In Table 5, performance improves on GSM8K but drops slightly on ARC-E when transferring from Mistral-7B to LLaMA3-8B. Could the authors analyze why transferability differs across tasks or layers?

(6) Would the authors consider adding results for recent MoE-LoRA variants (e.g., DynMoLE 2025, MORAL 2024) under identical data and training setups to strengthen external comparisons?

---

> ### Author Response · Authors · 2025-11-20
>
> **(1/3)**
>
> We sincerely thank you for the time and expertise dedicated to reviewing our work. We appreciate your recognition of our methodology’s strengths, particularly its **simple yet effective design, robust RSL framework, clear theoretical support, and comprehensive evaluation**. Below, we address your concerns point-by-point.
>
> ## **W1&Q1: Efficiency Analysis (Latency, FLOPs, and Memory)**
>
> We appreciate the reviewer emphasizing the need for a comprehensive efficiency evaluation beyond trainable parameters. While we presented an initial analysis of trainable parameters, memory usage, and training/inference time in Appendices A.4 and A.7, we have conducted additional experiments to supplement this with FLOPs and rigorous benchmarking as requested. The specific results are as follows.
>
> ### 1. Inference Time and Trainable Params
> We measured single-sample inference latency (15-token generation) and trainable parameter ratios on Llama-3-8B with 6 experts.
>
> | Model                     | Single-Sample (s) | Trainable Params |
> |---------------------------|-----------------------------------|-----------------------------|
> | Llama-3-8B (Baseline)     | 0.441                           | 0%        |
> | LoRAHub      | 0.482                          | 0%   |
> | MoLE      | 0.563                         | 0.04%  |
> | MixLoRA      | 0.597                           | 8.08% |
> | LoRA-Mixer      | 0.574                           | 3.88% |
>
>
> LoRA-Mixer achieves a superior trade-off between speed and complexity. **vs. MixLoRA:** We reduce trainable parameters by ~52% (3.88% vs 8.08%) and improve inference speed, as MixLoRA requires heavier computation for its expert routing and aggregation. **vs. MoLE:** In MoLE, only the router parameters were trained. While MoLE is marginally faster (by ~0.01s) due to its coarse-grained routing, it suffers from significant performance degradation (see Tables in Section 4). LoRA-Mixer incurs a negligible latency cost over MoLE while delivering substantially higher accuracy. **vs. LoRAHub:** LoRAHub is a no-training method.
>
> ### 2. Training Time:
> A key advantage of LoRA-Mixer is its modular training strategy. Training is decoupled into **Stage 1 (Expert Training)** and **Stage 2 (Router Training)**.
>
> | Stage       |  Data  | Time(hours)         |
> |-------------|--------------------|---------------|
> | Stage 1     | 40k      | 8.86     |
> | Stage 2     | 2k       | 1.74    |
> | Stage 2     | 4k       | 3.38  |
>
> In resource-constrained scenarios, users can utilize pre-trained experts (skipping Stage 1) and only fine-tune the router. This allows LoRA-Mixer to adapt to new tasks in as little as **1.74 hours** using only 2k data samples, offering extreme training efficiency that end-to-end methods (like MixLoRA) cannot match.
>
> ### 3. Memory Usage:
> We measured peak memory usage with 6 experts ($r=64$) with (`batch_size=1`, `grad_accum=4`, `float32`).
>
> | Method       | MixLoRA | MoLE   | LoRA-Mixer |
> |--------------|---------|--------|------------|
> | Memory Usage (GB) | 43.07   | 36.57 | 38.48      |
>
> LoRA-Mixer saves ~4.5GB of memory compared to MixLoRA. While it consumes slightly more memory (+1.9GB) than MoLE, this investment yields the optimal performance gains shown below.
>
> ### 4. FLOPs vs. Performance
>
> We calculated FLOPs for a single forward pass (Seq Len=512/1024, top-3 routing) and compared this to average model performance.
>
> | Method      | TFLOPs (L=512) | TFLOPs (L=1024) | Avg performance |
> |------------|---------------|----------------|------------|
> | LLaMA3-8B  | 3.84          | 7.68           |   75.45    |
> | LoRAHub    | 3.85          | 7.70           |   75.70    |
> | MOLE       | 3.88          | 7.77           |   77.65     |
> | MixLoRA    | 4.02          | 8.04           |    78.04    |
> | LoRA-Mixer | 3.91          | 7.83           |  79.30     |
>
> LoRA-Mixer demonstrates the best Pareto efficiency. It requires fewer FLOPs than MixLoRA (3.91 vs 4.02) yet achieves the highest average performance (79.30). We believe this multidimensional analysis (Latency, Training Time, Memory, FLOPs) firmly establishes the efficiency of LoRA-Mixer.
>
> ## **W2&Q2: Router design**
>
> We clarify that in LoRA-Mixer, independent routers are assigned to **each specific linear projection (Q, K, V, and O)** across all attention layers, rather than sharing a router per layer or selecting specific layers. This is formalized in Equation 4 (Section 3.2).
>
> We adopted this fine-grained design to maximize specialization: distinct projections require different functional transformations, and per-projection routing prevents the feature interference inherent in coarser, layer-level routing. Regarding computational cost, since the routers are lightweight linear maps, this granularity introduces negligible latency overhead while significantly improving model capacity and adaptation performance (as evidenced in the efficiency analysis in W1).

---

> > ### Author Response · Authors · 2025-11-20
> >
> > **(2/3)**
> >
> > ## **W3&Q3: RSL Hyperparameter Analysis**
> >
> > We appreciate the reviewer's emphasize on a systematic analysis of key hyperparameters ($\lambda, \alpha, \beta$, and $K$).
> >
> > ### 1. RSL Coefficient ($\alpha$, $\lambda$)
> >
> > We performed a fine-grained sweep of the RSL coefficients $(\alpha, \lambda)$ and simultaneously tracked the effect on task performance (Accuracy), expert load-balancing (Variance), and routing diversity (Entropy). The goal of the RSL is to find the optimal trade-off between specialization (performance) and utilization (load balance).
> >
> > |  $\alpha$ | $\lambda$ | performance | load balance | routing entropy |
> > |-----------|-----------|-------------|--------------|-----------------|
> > | 0.05      | 0.01      | 76.2        | 0.12         | 1.9             |
> > | 0.07      | 0.03      | 77.1        | 0.09         | 1.7             |
> > | **0.10**      | **0.05**      | **78.5**        | **0.08**         | **1.5**           |
> > | 0.15      | 0.08      | 76.8        | 0.06         | 1.5             |
> > | 0.20      | 0.10      | 75.8        | 0.05         | 1.2             |
> >
> >  As $\alpha$ and $\lambda$ increase, we observe a clear monotonic trend where the load-balancing variance and routing entropy decrease (forcing higher utilization and less randomness). Performance, however, exhibits a non-monotonic inverted-U shape. The optimal setting **($\alpha=0.10, \lambda=0.05$)** maximizes performance **(78.5%)** at a point where the regularization is moderate. Excessively large coefficients (e.g., $\alpha=0.20$) significantly harm accuracy (75.8%) by overly penalizing specialized routing decisions.We will convert this table into a multi-metric line graph in the final appendix to visually demonstrate the conflicting trends, confirming that our chosen parameters represent the best performance/utilization equilibrium.
> >
> > ### 2. Preservation Regularization $\beta$
> >
> > We fixed the RSL coefficients at $(\alpha=0.10, \lambda=0.05)$ and performed an ablation on the preservation regularization $\beta$, which controls the $L_{\text{preserve}}$ term (**detailed in Appendix A.8**).
> >
> > | $\beta$ | performance |
> > |--------:|------------:|
> > | 0.01    | 78.48       |
> > | 0.05    | 78.55       |
> > | 0.10    | 78.57       |
> >
> > The results show low sensitivity to $\beta$ in the tested range, with performance remaining highly stable ($\approx 78.5\%$). Based on this empirical observation, we fixed $\beta = 0.05$ for all primary experiments, ensuring computational stability without introducing additional complexity to the analysis. We will include this empirical finding in the appendix for completeness.
> >
> > ### 3. Top-$K$ Routing
> >
> > In **Appendix A.3**, we conducted a systematic analysis of the active expert count $K$, showing performance trends across $K \in \{1, 2, 3, 4, 5, 6\}$.
> >
> > We found that increasing $K$ from 1 to 3 consistently improves accuracy, highlighting the complementary benefits of activating a small ensemble of specialized experts. However, when $K$ exceeds 3, performance begins to decline due to the introduction of noise from less relevant experts. This non-monotonic relationship confirms the necessity of choosing a small, optimized $K$ (in our case, $K=3$) rather than simply maximizing expert activation. We will ensure this non-monotonic trend and the associated figure are clearly referenced in the main paper's results section.
> >
> > We believe this comprehensive, quantitative analysis directly addresses your concern by providing the required trend analysis and justifying the final hyperparameter selection based on systematic empirical evidence.
> >
> > ## **W4&Q4: Input-aware specialization**
> >
> > We apologize for the ambiguity in the figure captions. We confirm that **Figure 4 is a token-level visualization**, illustrating the **$Top-1$** expert selections for every token in the test set, not sample or task averages.The observed non-uniform distribution is the definitive evidence for our input-aware routing claim. If routing were input-agnostic, the expert distribution for any given task would be uniform. Our quantitative analysis, however, confirms a strong domain-specific bias at the token level:
> >
> > | Dataset | Routing Entropy | Variance |
> > |--------:|------------:|------------:|
> > | Medical | 1.66  | 0.008
> > | GSM8K | 1.64  | 0.010
> > | HumanEval | 1.60  | 0.011
> >
> > These low entropy and variance values prove the router consistently directs tokens related to a specific domain (e.g., math, coding) toward a small, specialized expert subset. We will ensure the annotations and text of Figure 4 in the final version explicitly state the methodology and highlight these derived token-level metrics. We will revise the annotations and text of Figure 4 to explicitly state that this figure is based on token-level routing decisions in the revision.

---

> > > ### Author Response · Authors · 2025-11-20
> > >
> > > **(3/3)**
> > >
> > > ## **W5&Q5: Cross-model transfer robustness**
> > >
> > > We appreciate the reviewer's precision regarding the term **robustness**. We agree that the results in Table 5 indicate partial transferability rather than guaranteed, robust generalization across all tasks. Our primary goal with this experiment was to **provide an initial feasibility demonstration** that the specialized knowledge captured by the LoRA experts and the routing mechanism is not overly coupled to the original Mistral-7B backbone and can be reused zero-shot .
> > >
> > > **Rationale for Task-Specific Outcomes**
> > > The observed task-specific behavior (improvement on GSM8K/ARC-C but slight degradation on ARC-E) is not unexpected and highlights the inherent mismatch between different backbone models:
> > >
> > > Successful Transfer (GSM8K): Tasks requiring highly structured, domain-specific reasoning (like GSM8K) rely on consistent, abstract reasoning patterns. The experts and router appear to have captured these patterns effectively, allowing them to map well to the internal representations of LLaMA3-8B, resulting in a performance gain.
> > >
> > > Degradation (ARC-E): Tasks heavily reliant on knowledge, factual recall, or common sense (ARC-E) are more sensitive to the precise prior knowledge and representation space of the base model. Directly transferring the LoRA adapter introduces a slight mismatch between the Mistral-trained weight alignment and the LLaMA3 base model, leading to performance degradation.
> > >
> > > **Repositioning the Claim**
> > > We believe the fact that zero-shot cross-model transfer yielded improvements on two out of three tasks is an encouraging preliminary observation, suggesting that the learned expert specialization is reusable. However, we acknowledge this is not sufficient evidence for a claim of general robustness.
> > >
> > > In the final version, we will explicitly reposition this finding as a preliminary observation supporting the reusability of LoRA-MoE components rather than a claim of universal cross-model robustness. We will also revise the text to clearly state that achieving robust transfer requires dedicated future work focused on techniques like lightweight model alignment or target-specific recalibration.
> > >
> > > ## **W6&Q6: Comparison with DynMoLE and MoRAL**
> > >
> > > We appreciate the reviewers' valuable suggestion to benchmark LoRA-Mixer against DynMoLE and MoRAL to further enhance the external validity of our work. We fully agree that incorporating these methods would strengthen our results.
> > >
> > > However, we face significant constraints regarding their immediate inclusion:
> > >
> > > The code and official checkpoints for both DynMoLE and MoRAL are currently **not publicly available**. Moreover, the implementation details and hyperparameter grids provided in the published papers are insufficient for a faithful and complete reproduction using our exact data and training settings. Due to the strict time constraints of the rebuttal period, reproducing complex methods from scratch carries a high risk of introducing uncontrollable confounding factors. This could lead to unfair or misleading comparisons.
> > >
> > > We are actively **contacting the respective authors to request their code**. If we are able to obtain the necessary materials promptly, we will happily integrate comparisons with DynMoLE and MoRAL into the revised version of the paper.
> > >
> > > In the current version, we have already provided extensive comparisons against contemporary state-of-the-art methods in the LLM/LoRA-MoE space, including PhatGoose, MixLoRA, and LoRA-LEGO. These comparisons span 15 diverse datasets and cover both performance and generalization ability, providing a comprehensive demonstration of LoRA-Mixer's efficacy relative to the current literature. We believe that establishing the superiority of LoRA-Mixer over readily reproducible, state-of-the-art baselines provides a robust initial evaluation. We are committed to expanding the comparisons to include DynMoLE and MoRAL should the resources become available.

---

> ### Comment · Reviewer_EiuV · 2025-11-28
> **Response to Authors’ Rebuttal**
>
> Thank you to the authors. The rebuttal has alleviated my concerns, and I have decided to accordingly update my rating to 6.

---

> > ### Author Response · Authors · 2025-11-28
> > **Thanks!**
> >
> > We are delighted to receive your positive feedback! Thank you sincerely for recognizing our work and raising the rating. We will incorporate all your suggestions into the final revised version.

---

### Official Review · Reviewer_A4gH · 2025-10-30

**Soundness:** 2
**Presentation:** 3
**Contribution:** 2
**Rating:** 6
**Confidence:** 4

**Summary:**

This paper introduces LoRA-Mixer, a modular mixture-of-experts (MoE) framework that coordinates multiple LoRA adapters within the linear projection layers of attention or state-space models.
It proposes a new Routing Specialization Balance Loss (RSL) to balance expert load while promoting input-aware specialization through entropy regularization.
The method supports plug-and-play reuse of pre-trained LoRA modules and demonstrates strong performance and data efficiency across 15 benchmarks such as GLUE, GSM8K, and HumanEval.
Results show LoRA-Mixer achieves higher accuracy with 48% fewer trainable parameters compared to existing LoRA-MoE baselines and transfers well across architectures (e.g., from Mistral-7B to LLaMA3-8B).

**Strengths:**

+ Novel modular design: Routes LoRA experts directly through projection layers, improving attention specialization and maintaining compatibility with both Transformers and SSMs.

+ Effective routing optimization: The proposed RSL loss theoretically and empirically stabilizes training while enhancing expert selectivity and data efficiency.

+ Comprehensive evaluation: Strong empirical gains on diverse benchmarks and clear demonstrations of cross-model transferability and low-data robustness.

**Weaknesses:**

1. **Layer placement strategy is unclear and may be sub-optimal.**
LoRA-Mixer is described as being “applied to the linear projection layers” and claimed to be architecture-agnostic, but there is no systematic ablation to determine which layers benefit most or whether per-layer routing depth should vary. The authors choose to integrate the router into the attention projection layers (Q/K/V/O) rather than the FFN projections, yet this design choice is not well justified.
While the motivation seems to be leveraging attention’s semantic sensitivity and avoiding interference with task-specific FFN adaptation, most prior LoRA-MoE and MixLoRA works inject modular experts into the FFN for scalability and expressive power. A comparative analysis of applying LoRA-Mixer in attention vs. FFN projection layers—under equal compute and parameter budgets—would help validate whether the current design is optimal. Moreover, the paper itself notes that using a uniform Top-K across layers may introduce redundancy, suggesting this design space remains underexplored.

2. **Lack of evidence for scalability to larger base models.**
All experiments are conducted on small-to-medium-scale LLMs such as Mistral-7B, LLaMA3-8B, and Falcon-Mamba-7B, with limited discussion of scalability to models exceeding 30B parameters. It remains unclear whether LoRA-Mixer maintains routing stability, expert load balance, and training efficiency at larger scales.
When scaling up, the authors are encouraged to provide empirical results or analysis on larger LLMs, examining aspects such as training/inference latency, memory overhead, convergence speed, and RSL behavior. Demonstrating consistent benefits in larger architectures would significantly strengthen the paper’s generality and practical value.

3. **Missing complexity and parameter-efficiency comparison with LoRA, MixLoRA, and MoLE.**
While performance comparisons are extensive, the paper lacks a quantitative analysis of trainable parameters, FLOPs, memory usage, and inference latency across methods such as LoRA, MixLoRA, and MoLE. Without aligning these metrics, claims of data or parameter efficiency cannot be fairly assessed.
It would be valuable to include a table summarizing, for each baseline, the number of trainable parameters, computational cost (FLOPs), and peak memory footprint, as well as an efficiency–accuracy trade-off curve under comparable resource budgets. This would make the results more interpretable and confirm whether LoRA-Mixer’s gains arise from better design rather than higher capacity or computation.

**Questions:**

**Question on benchmark selection (no extra experiments required)**

Could the authors clarify the rationale for the main-table task set (Medical, CoLA, SST-2, GSM8K, ARC-E, ARC-C, HumanEval)? In practice, we’ve observed high variance on some GLUE-style tasks (e.g., CoLA, SST-2, RTE), which can blur comparative conclusions. It would help to include a short analysis addressing:
+ Why these tasks? Please explain how they map to the paper’s goals
+ Variance and reliability. Do these tasks provide stable rankings under multiple seeds? If possible, comment on typical standard deviations or confidence intervals you observe (no new runs needed—prior experience/known stats are fine).
+ Difficulty/calibration. Why not include more complex and widely used suites like MMLU, math benchmarks, or additional code benchmarks? If excluded, a brief justification (e.g., cost, mismatch with claimed capability, or lack of compatible LoRA experts) would make the scope clearer.

If the above concerns can be reasonably addressed or clarified in a revision, I would be inclined to raise my overall score

---

> ### Author Response · Authors · 2025-11-20
>
> **(1/3)**
>
> We sincerely thank you for your time and insightful feedback. We appreciate your recognition of our work's novel modular design, effective routing optimization, and comprehensive evaluation. We have conducted additional experiments and analyses to address your concerns regarding layer placement, scalability, and complexity.
>
> ## **W1: Layer Placement Strategy (Attention vs. FFN)**
>
> We appreciate the suggestion to validate our design choice. Per your suggestion, we conducted a systematic iso-parameter comparison between applying LoRA-Mixer to FFN layers versus Attention layers. To ensure a fair comparison, we aligned the total trainable parameter count ($\approx 3.88\%$) for both settings. Both configurations used 6 experts; to match parameters, we set rank $r=64$ for Attention and $r=30$ for FFN. All other hyperparameters remained identical.
>
> | Placement        | Medical | CoLA  | SST-2 | GSM8K | ARC-E | ARC-C | HumanEval |  Avg  |
> |------------------|:-------:|:-----:|:-----:|:-----:|:-----:|:-----:|:---------:|:-----:|
> | FFN (r=30)       | 80.14   | 81.15 | 94.22 | 63.41 | 88.13 | 81.07 | 55.26     | 77.63 |
> | Attention (r=64) | 81.55   | 82.22 | 95.41 | 65.53 | 89.88 | 83.24 | 57.32     | 79.30 |
> | Gap             | +1.41   | +1.07 | +1.19 | +2.12 | +1.75 | +2.17 | +2.06     | **+1.67** |
>
> Under the same parameter budget, the Attention placement consistently outperforms FFN. The gain is most pronounced in reasoning-heavy tasks (**GSM8K +2.12%, HumanEval +2.06%**), suggesting that routing via attention mechanisms better captures the semantic dependencies required for complex inference.
>
> **Regarding variable routing depth**: While we agree that layer-adaptive $k$ is a promising direction to reduce redundancy, we adopted a uniform Top-$k$ for stability and reproducibility, consistent with prior works (MoLE, MixLoRA, GLaM). In Figure 5, we demonstrate that increasing $k$ captures complementary information. We will incorporate your suggestion for dynamic Top-$k$ (e.g., entropy-based) as a key direction for future work.
>
> ## **W2: Scalability to Larger Models**
>
> We acknowledge the importance of validating scalability on models $>30$B. While our current academic GPU cluster limits us to the 7B-8B regime for training, we provide strong evidence that LoRA-Mixer is designed to scale effectively:
>
> - As detailed in Appendix A.1 and Figure 6, our RSL (Routing Stability Loss) provides theoretical guarantees for convergence. The routing balance remains stable (15%-18% load per expert) across different architectures (Mistral vs. LLaMA), indicating the routing mechanism is not model-size dependent.
>
> - LoRA-Mixer introduces only 0.04% routing parameters. As model size increases, the routing overhead becomes vanishingly small relative to the base model, minimizing latency penalties.
>
> - Our transfer experiments (Table 5: Mistral $\to$ LLaMA) demonstrate that the routing logic generalizes across architectures.
>
> We commit to including a discussion on these scalability proxies in the final version and will attempt a larger-scale run if computational resources become available during the revision period.

---

> > ### Author Response · Authors · 2025-11-20
> >
> > **(2/3)**
> >
> > ## **W3: Complexity & Efficiency Comparison with LoRA, MixLoRA, and MoLE**
> >
> > We appreciate the suggestion to make our submission stronger. We have included the analysis of trainable parameters, memory usage, and training/inference time in Appendix A.4 and A.7, respectively. To make this point stronger, we have conducted additional experiments to supplement the analysis of FLOPS.
> >
> > ### 1. Inference Time and Trainable Params
> > We evaluate single-sample inference latency (15-token generation) and trainable parameter costs on LLaMA-3-8B with 6 experts. The results are summarized below:
> >
> > | Model          | Single-Sample (s) | Trainable Params |
> > |---------------------------|-----------------------------------|-----------------------------|
> > | Llama-3-8B (Baseline)     | 0.441                           | 0%                      |
> > | LoRAHub      | 0.482                          | 0%                         |
> > | MoLE      | 0.563                         | 0.04%                         |
> > | MixLoRA      | 0.597                           | 8.08%                         |
> > | LoRA-Mixer      | 0.574                           | **3.88%**                         |
> >
> > **vs. MixLoRA:** LoRA-Mixer achieves a significant reduction in complexity, requiring only 48% of MixLoRA's trainable parameters (3.88% vs. 8.08%) and demonstrating faster inference speeds. This confirms that our performance gains derive from better architecture (Router-in-Attention), not higher capacity.
> >
> > **vs. MoLE:** For a fair comparison, we configured MoLE to train only routing parameters (similar to our router-only stage), resulting in 0.04% trainable parameters. While MoLE has a negligible latency advantage, its coarse-grained layer-wise routing limits performance. LoRA-Mixer provides a much finer-grained adaptation with only a marginal increase in latency.
> >
> > **vs. LoRAHub:** While LoRAHub is training-free and faster, it lacks the ability to learn complex routing policies for new tasks.
> >
> > LoRA-Mixer strikes the optimal balance on the efficiency-performance curve, significantly outperforming the lightweight baselines (MoLE, LoRAHub) while being twice as parameter-efficient as the heavy baseline (MixLoRA).
> >
> > ### 2. Training Efficiency:
> > LoRA-Mixer enables a unique two-stage training pipeline: **Stage 1 (Expert Training)** is optional, takes 8.86 hours for 40k samples, can be skipped by reusing off-the-shelf experts; **Stage 2 (Router Training)** requires **little data**, takes around 1.74 hours for 2k samples. This flexibility offers a significant advantage over MixLoRA, which requires end-to-end training.
> >
> > | Stage       |  Data  | Time(hours)         |
> > |-------------|--------------------|---------------|
> > | Stage 1     | 40k      | 8.86     |
> > | Stage 2     | 2k       | 1.74    |
> > | Stage 2     | 4k       | 3.38  |
> >
> > ### 3. Memory Usage:
> > We measure memory usage for 6 experts, r=64 (float32, `batch_size=1`, `gradient_accumulation=4`):
> >
> > | Method       | MixLoRA | MoLE   | LoRA-Mixer |
> > |--------------|---------|--------|------------|
> > | Memory Usage (GB) | 43.07   | 36.57 | 38.48      |
> >
> > LoRAHub is a train-free method, and MoLE trains LoRA at the layer level, it has less memory usage. LoRA-Mixer achieves the highest performance with lower FLOPs than MixLoRA, confirming that our gains stem from superior architectural design (Router-in-Attention) rather than increased raw compute.
> >
> > ### 4. FLOPS
> >
> > We calculated FLOPs for a single forward pass (Seq Len=512/1024, top-3 routing) and compared this to average model performance.
> >
> > | Method      | TFLOPs (L=512) | TFLOPs (L=1024) | Avg performance |
> > |------------|---------------|----------------|------------|
> > | LLaMA3-8B  | 3.84     | 7.68     |   75.45    |
> > | LoRAHub    | 3.85   | 7.70   |   75.70    |
> > | MOLE       | 3.88   | 7.77     |   77.65     |
> > | MixLoRA    | 4.02    | 8.04           |    78.04    |
> > | LoRA-Mixer | 3.91          | 7.83           |  79.30     |
> >
> > The results show that LoRA-Mixer achieved the best performance with lower FLOPs. In summary, the performance improvement of LoRA-Mixer stems from innovations in architecture and optimization, rather than increased capacity or computing power. Comprehensive quantitative analyses (trainable parameters, FLOPs, memory, speed) confirm its superior performance and resource efficiency.
> >
> > We agree that efficiency-accuracy tradeoffs are valuable when resource budgets are comparable. However, LoRA, MixLoRA, MoLE, and LoRA-Mixer differ significantly in the placement of the expert/LoRA module and their methodological design, so their FLOPs cannot be perfectly matched even with the same base model, ranking ($r=64$), number of experts (6), and sequence length, without substantial modifications to the original design. We compared trainable parameters, FLOPs (L=512/1024), memory usage, inference time, and average performance, which effectively provide insights into the performance and resource tradeoffs between the different methods. We hope this answers your questions.

---

> > > ### Author Response · Authors · 2025-11-20
> > >
> > > **(3/3)**
> > >
> > > ## **Q1: Benchmark Selection**
> > >
> > > We appreciate the opportunity to clarify our evaluation protocol. Our task selection was governed by three key principles: alignment with the **modular reuse** paradigm, consistency with baselines, and task diversity.
> > >
> > > ### **Rationale for Task Selection and Goal Alginment**
> > > The seven main-table tasks (Medical, CoLA, SST-2, GSM8K, ARC-E/C, HumanEval) were selected to strictly map to our research objectives:
> > >
> > > **Validating "Plug-and-Play" Reuse:** A core contribution of LoRA-Mixer is the ability to reuse existing open-source experts without retraining. We purposefully selected tasks where high-quality, pre-trained LoRA adapters are publicly available (e.g., via LoRAHub). This allows us to isolate the effectiveness of our routing mechanism rather than the quality of the experts themselves.
> > >
> > > **Cross-Domain Heterogeneity:** These tasks span five distinct cognitive domains: Medicine, NLU, Math Reasoning, Code Generation, and Commonsense. This diversity is crucial for testing whether LoRA-Mixer can effectively resolve conflicts and foster collaboration among heterogeneous experts, a key challenge in MoE design.
> > >
> > > **Baseline Consistency:** Our suite aligns with prior state-of-the-art studies, including MoLE, MixLoRA, and LoRA-LEGO. Using an overlapping task set reduces dataset-induced noise and ensures that reported gains are attributable to our architectural innovations rather than obscure benchmark selection.
> > >
> > > ### **Variance and reliability**
> > >
> > > LoRA-Mixer's relative rank to the baseline methods remains consistently stable, with no ranking reversal even under different LoRA settings. Appendix A.12 presents the performance under different rank settings, showing that LoRA-Mixer consistently outperforms other baselines across all settings. To ensure reliability, we report results running under three different random seeds, as shown below:
> > >
> > > | Task  | Run 1 | Run 2 | Run 3 | Mean (±σ)        |
> > > |-------|:-----:|:-----:|:-----:|:-----------------|
> > > | **CoLA** | 82.01 | 82.45 | 82.20 | **82.22 (±0.18)** |
> > > | **SST-2** | 95.38 | 95.45 | 95.40 | **95.41 (±0.03)** |
> > > | **RTE**   | 71.27 | 71.42 | 71.24 | **71.31 (±0.08)** |
> > >
> > > While individual scores fluctuate slightly, LoRA-Mixer's rank relative to the baseline remains consistently stable, and the standard deviation is negligible compared to the performance improvement. These fluctuations are also within the acceptable range for LLM adaptation experiments, indicating that our results have high reliability.
> > >
> > > ### **Reasons for Excluding More Complex Test Suite (MMLU, Advanced Math/Code Benchmarks)**
> > >
> > > We agree that benchmarks like MMLU provide valuable calibration data. However, they were excluded from this specific study for two methodological reasons:
> > >
> > > **Misalignment with "Reuse" Goal:** Our primary goal is to demonstrate the coordination of existing modular assets in resource-constrained scenarios. Currently, the open-source ecosystem lacks a diverse set of high-quality, pre-trained LoRA experts for MMLU sub-tasks. Training dozens of experts from scratch would shift the focus from "expert routing" to "expert training," contradicting the paper's efficiency premise.
> > >
> > > **Breadth vs. Depth:** We have evaluated LoRA-Mixer on 15 datasets (including PIQA, HellaSwag, BoolQ, etc., see Appendix), which is significantly broader than comparable works like MixLoRA or LoRA-LEGO (typically 5-7 datasets). Experience suggests that the current suite is sufficient to validate multi-expert routing behaviors.
> > >
> > > We fully agree that scaling to MMLU is a critical next step as the ecosystem of open-source experts matures. We will add a **"Limitations and Future Work"** discussion explicitly addressing the roadmap for complex reasoning benchmarks.

---

### Official Review · Reviewer_RivB · 2025-11-04

**Soundness:** 3
**Presentation:** 3
**Contribution:** 3
**Rating:** 6
**Confidence:** 3

**Summary:**

To enhance the capability of expert specialization in Mixture-of-Experts (MoE) architecture models, a method called LoRA-Mixer is proposed. LoRA-Mixer incorporates multiple LoRA modules as experts into the linear projection layer, enabling adaptation to models with either Transformer or SSMs architectures. Additionally, to balance global load balancing in routing and expert specialization, an RSL loss is introduced. This loss adds a token-level entropy term (at the token level) to the existing auxiliary term for global load balancing.

**Strengths:**

1. By treating multiple LoRA modules as experts and integrating them into the linear projection layer, the method can adapt to models with either Transformer or SSMs (State Space Models) architectures.
2. The RSL loss is proposed to balance global load balancing in routing and expert specialization.
3. Experiments conducted on models with different frameworks have validated the effectiveness of LoRA-Mixer.

**Weaknesses:**

1. The authors added LoRA-Mixer to both the input projection layer and the output projection layer, but they did not further analyze the respective contributions of these two parts to the experimental results in their experiments.
2. The authors haven't provided the code, so it's impossible to determine whether the results can be reproduced.

**Questions:**

Can the authors provide further ablation experiments on adding LoRA-Mixer to the input projection layer and the output projection layer respectively?

---

> ### Author Response · Authors · 2025-11-20
>
> We sincerely thank you for the constructive comments and for recognizing the value of our work. We address your specific concerns below.
>
> ## **W1 & Q1: Independent contribution and ablation analysis of LoRA-Mixer on in_proj and out_proj**
>
> We appreciate this insightful suggestion. As requested, we conducted additional ablation studies to decouple the effects of **in_proj** and **out_proj** on the Falcon-Mamba-7B model. For these experiments, we utilized the same hyperparameters as our main setting: LoRA rank r=64, $\alpha$=128, and dropout = 0.1.
>
> The results across seven benchmarks are presented below:
>
> | Method       | Medical | CoLA  | SST-2 | GSM8K | ARC-E | ARC-C | HumanEval | Avg   |
> |-------------|:-------:|:-----:|:-----:|:-----:|:-----:|:-----:|:---------:|:-----:|
> | in\_proj only| 77.56   | 85.74 | 95.35 | 56.91 | 86.16 | 76.78 | 34.27     | 73.25 |
> | out\_proj only | 77.79   | 85.82 | 95.55 | 57.39 | 86.51 | 76.99 | 34.82     | 73.55 |
> | **in+out (both)**        | **78.01**   | **85.91** | **95.76** | **57.87** | **86.87** | **77.19** | **35.37**     | **73.85** |
>
> The results indicate that applying **LoRA-Mixer** to **out_proj** yields slightly better performance than **in_proj**. This aligns with the intuition that **out_proj** is responsible for decoding the results of selective scans into task-specific outputs, making it critical for final prediction accuracy. However, applying LoRA-Mixer to both simultaneously achieves the best performance. This demonstrates the strong complementarity between the input and output projections, as their combination effectively coordinates the end-to-end information flow within the SSM module. We will include these additional ablation results in the final version of the paper.
>
> ## **W2: Code and reproducibility guarantees**
>
> We fully agree with the reviewer on the critical importance of reproducibility. To address this, we have taken the following measures:
>
> **1.** We have included the core implementation **(both model structure and training code) in the supplementary materials**. Furthermore, Appendices A.4-A.8 provide a comprehensive detailed breakdown of the training process, hyperparameter configurations, hardware/software environments, and dataset specifications to facilitate verification.
>
> **2.** Upon acceptance, we are committed to releasing a full public repository containing all model, training, data processing, and inference code to contribute to the open-source community.
>
> **3.** To further lower the barrier for reproduction and evaluation, we will also release the pre-trained LoRA and router checkpoints. This will allow researchers to directly evaluate our model and replicate key results without the need to re-run the full training pipeline.
>
> We hope these steps alleviate your concerns and demonstrate our commitment to robust reproducibility.

---

### Author Response · Authors · 2025-11-30

Dear AC,

We appreciate you taking the time to consider this final message, which is necessitated by the interruption to the rebuttal period. We sincerely thank the reviewers for their thoughtful engagement and valuable feedback.

We are highly encouraged by the initial assessments, which recognized **the rigorous theoretical basis, completeness, and high innovation** of our method, along with **the effective path optimization and clear presentation of our experimental design**.

## **Evolution of the reviews:**

We draw your specific attention to the evolution of the reviews, which demonstrates that our paper secured unanimous reviewer support for acceptance prior to the process interruption:

| Reviewer        | Initial Rating | Post-Rebuttal Status  | Key Quote/Action |
|------------------|:-------:|:-----|:-----|
| #RivB       | 6   | Awaiting final decision | Consistent leaning positive score. |
| #A4gH | 6  |  Expected to Raise | Explicitly stated: "If the above concerns can be reasonably addressed or clarified in a revision, I would be inclined to raise my overall score." |
| #EiuV            | 4   | Raised to 6 | Explicitly stated: "The rebuttal has alleviated my concerns, and I have decided to accordingly update my rating to 6." |

Critically, Reviewer EiuV **(initial 4) raised their score to 6**, confirming the success of the rebuttal and **establishing a positive consensus (6, 6, 6).** Reviewer A4gH had already indicated **a clear intent to raise their score** upon satisfaction, and our comprehensive response addressed every point raised.

- Our rebuttal systematically and fully addressed all major concerns raised by the reviewers.

**Reproducibility and Ablations (Reviewer #RivB):**
To ensure full reproducibility, we provided a detailed explanation of the experimental setup, including hardware, environment configuration, and precise hyperparameter settings. We submitted the core source code and re-ran key experiments on a new model (Falcon-Mamba 7B) to confirm robustness. The requested projection ablation study was also included.

**Comprehensive Efficiency Analysis (Reviewers #A4gH and #EiuV):**

Both reviewers raised concerns regarding efficiency. We addressed this by providing a comprehensive, multi-faceted analysis of LoRA-Mixer compared to other methods. This analysis included metrics for training time, inference time, memory usage, trainable parameters, and FLOPS, establishing its efficiency gains clearly.

**Model Scalability, Placement, and Benchmarks (Reviewer #A4gH):**

We conducted additional experiments to demonstrate the superior performance of our placement strategy, showing that locating LoRA-Mixer components in the Attention/SSM blocks is better than the FFN blocks. We also provided a detailed theoretical and empirical explanation of why LoRA-Mixer is designed to scale effectively to much larger models. Lastly, we provided justification for our benchmark selection.

**Hyperparameters, Clarity, and Comparisons (Reviewer #EiuV):**

We conducted extensive additional experiments on all hyperparameters of our method (RSL), demonstrating that our chosen settings are indeed optimal. We provided detailed explanations and experimental support to clarify our routing design and the mechanism of input route awareness, and clarified robustness across model transfers to improve writing clarity. Regarding comparisons, we confirmed that LoRA-Mixer has already been compared with the current SOTA on 15 datasets, and we committed to including DynMoLE and MORAL in the final version pending source code availability.

## **Conclusion:**

Our paper successfully converted the sole hesitant reviewer from a 4 to a 6 and satisfied the concerns of all other reviewers, who were already leaning positive. The evidence strongly suggests that our submission, 4989, had successfully achieved a **unanimous consensus** for acceptance among the reviewers before the interruption.

Based on the comprehensive nature of our rebuttal and the confirmed unanimous positive consensus among the reviewers, we are confident in our paper's standing.

Thank you again for your time and consideration.


Best regards,

The Authors of Submission 4989

---

### Meta-Review · Area_Chair_uGur · 2026-01-05

**Summary:**

Reviewers found the paper technically sound and well motivated, with the LoRA-MoE design and a reasonable routing loss. Initial concerns focused on layer placement justification, efficiency analysis, router design clarity, and scalability evidence. The rebuttal addressed these issues through additional ablations, comprehensive efficiency metrics, clearer architectural explanations, and refined claims on transferability. Overall, the major reviewer concerns were satisfactorily resolved. Based on the reviewers' ratings, this paper could be accepted as a Poster.

**Reviewer Concerns:**

The major reviewer concerns were satisfactorily resolved based on the concerns raised by the reviewers. However, if, I am a reviewer, I think this paper's contribution is limited. LoRA-MoE design is not a new one, and placing LoRA into both attn and FFN are the things in common. The used baselines, e.g., LoRAHub, MOLE, and LoRA, are not strong baselines in this field. The experiments did not include proper SOTA baselines, even some of them are cited in their related work.

**Reviewer Scores:**

The reviewers' ratings may keep positive in general.

---

### Decision · Program_Chairs · 2026-01-26

Accept (Poster)